behaviour/ecology/physiology

niche divergence, feeding, fitness, marine mammal, spatial ecology, survival

**Author for correspondence:**
Sarah S. Kienle
e-mail: sarah_kienle@baylor.edu

# Trade-offs between foraging reward and mortality risk drive sex-specific foraging strategies in sexually dimorphic northern elephant seals

Sarah S. Kienle[1,3], Ari S. Friedlaender[2],
Daniel E. Crocker[4], Rita S. Mehta[1] and Daniel P. Costa[1]

[1]Ecology and Evolutionary Biology, and [2]Ocean Science, University of California, 130 McAllister Way, Santa Cruz, CA 95060, USA
[3]Department of Biology, Baylor University, One Bear Place #97399, Waco, TX 76798, USA
[4]Biology, Sonoma State University, 1801 East Cotati Avenue, Rohnert Park, CA 94928, USA

SSK, 0000-0002-8565-2870; ASF, 0000-0002-2822-233X;
DPC, 0000-0002-0233-5782

Sex-specific phenotypic differences are widespread throughout the animal kingdom. Reproductive advantages provided by trait differences come at a cost. Here, we link sex-specific foraging strategies to trade-offs between foraging reward and mortality risk in sexually dimorphic northern elephant seals (*Mirounga angustirostris*). We analyse a decadal dataset on movement patterns, dive behaviour, foraging success and mortality rates. Females are deep-diving predators in open ocean habitats. Males are shallow-diving benthic predators in continental shelf habitats. Males gain six times more mass and acquire energy 4.1 times faster than females. High foraging success comes with a high mortality rate. Males are six times more likely to die than females. These foraging strategies and trade-offs are related to different energy demands and life-history strategies. Males use a foraging strategy with a high mortality risk to attain large body sizes necessary to compete for females, as only a fraction of the largest males ever mate. Females use a foraging strategy with a lower mortality risk, maximizing reproductive success by pupping annually over a long lifespan. Our results highlight how sex-specific traits can drive disparity in mortality rates and expand species' niche space. Further, trade-offs between foraging rewards and mortality risk can differentially affect each sex's ability to maximize fitness.

# 1. Introduction

From the peacock's ornate tail to the male lion's distinctive mane, sex-specific differences abound across the animal kingdom. Iconic traits associated with sexual dimorphism go hand-in-hand with differences in behaviour, ecology, life history and physiology [1–3]. Extravagant and extreme sexually dimorphic traits can provide intrasexual advantages in reproductive success, but not without cost [4,5]. Understanding sex-specific life-history trade-offs is a central theme in evolutionary biology [6–8]. However, it is difficult to examine the fitness consequences of sex-specific behavioural and life-history strategies of wild animals due to logistical challenges (e.g. animal size, mobility, lack of accessibility) in obtaining these data from free-ranging populations [8,9]. Here, we link sex-specific foraging strategies that result in intraspecific niche divergence to trade-offs between foraging success and mortality rate in a sexually dimorphic wild mammal.

Animals strive to maximize energy intake relative to time or energy spent foraging [10,11]. Among sexually dimorphic taxa, the larger sex often requires additional or different prey resources, and/or the sexes exhibit different morphologies that affect feeding behaviour [12–14]. These biological differences can result in sex-specific foraging strategies—specific combinations of behaviour, morphology and physiology [11]—that impact reproductive success and survival [15–17]. Sex-specific foraging strategies can therefore intensify trait differences and play an essential role in the evolution and maintenance of sexual dimorphism [11,18,19].

Northern elephant seals (*Mirounga angustirostris*) are a classic example of sexual dimorphism. Adult males are three to seven times larger than females and have secondary sexual characteristics, including the long proboscis that inspired the species' common name [20,21]. The northern elephant seal's annual life cycle is divided between land and sea. Adults are only on land twice a year to breed (one to three months) and moult (approx. one month). For the rest of the year (8–10 months), adults are at sea on extended post-breeding and post-moult foraging trips.

Northern elephant seals fulfil many of the predictions associated with sexual selection. The species is highly polygynous [22]. Males fight conspecifics to establish harems to control access to mating opportunities and operate in one of the most competitive breeding systems on the planet [20,21,23]. Extreme size and strength are prerequisites for male reproductive success, with only a handful of males ever mating. Most females, however, breed every 1–2 years and have high reproductive success. Only females provide parental care [24,25].

Behavioural observations suggest that sexual dimorphism in northern elephant seals extends to niche divergence when feeding. Previous studies classified males as benthic continental shelf predators and females as deep-diving pelagic predators [23,26–28]. Spatial segregation of foraging habitat was hypothesized to be caused by differences in prey resources. Specifically, continental shelf habitats were thought to provide better prey resources necessary for sustaining males' larger body size [28]. Over the last two decades, a substantial increase in tracking data suggests that some females may undertake benthic foraging dives on or near the coast; these benthic females appear to have greater mass gain than pelagic-foraging females, similar to males [29–32]. These data question the dichotomy between male and female foraging strategies. It is now unclear whether sex-specific traits translate to niche divergence or whether habitat preference determines foraging strategy.

With more than 239 000 animals in this large, long-lived species, northern elephant seals exert a tremendous influence on coastal and open ocean ecosystems [28,33]. It is important to understand the variables that contribute to intraspecific differences in foraging strategies, as they influence the species' ecological niche space. Furthermore, different foraging strategies reflect cost–benefit trade-offs when feeding, thereby affecting individual fitness.

Long-term data from northern elephant seals provide the opportunity to examine the fitness consequences of different foraging strategies in sexually dimorphic animals. This study's first objective is to determine which factors lead to divergence in northern elephant seal foraging strategies. We integrate a decadal dataset on both sexes' spatial patterns and dive behaviour to test the hypothesis that seals exhibit foraging strategies that are based on both sex and habitat. The second objective is to assess trade-offs between alternative foraging strategies. We test whether foraging strategies represent a compromise between foraging success (reward) and mortality rate (risk). Mortality events are difficult to detect using satellite transmitted data, especially in wide-ranging animals that die in remote locations [34]. Here, we combine the satellite tracking dataset with long-term demographic data to overcome this challenge. Our results provide evidence that niche divergence contributes to the continued maintenance of sexual dimorphism in this species. Sex-specific traits can drive disparity in foraging success and mortality rates, expand species' overall niche space and result in different selective pressures to maximize fitness.

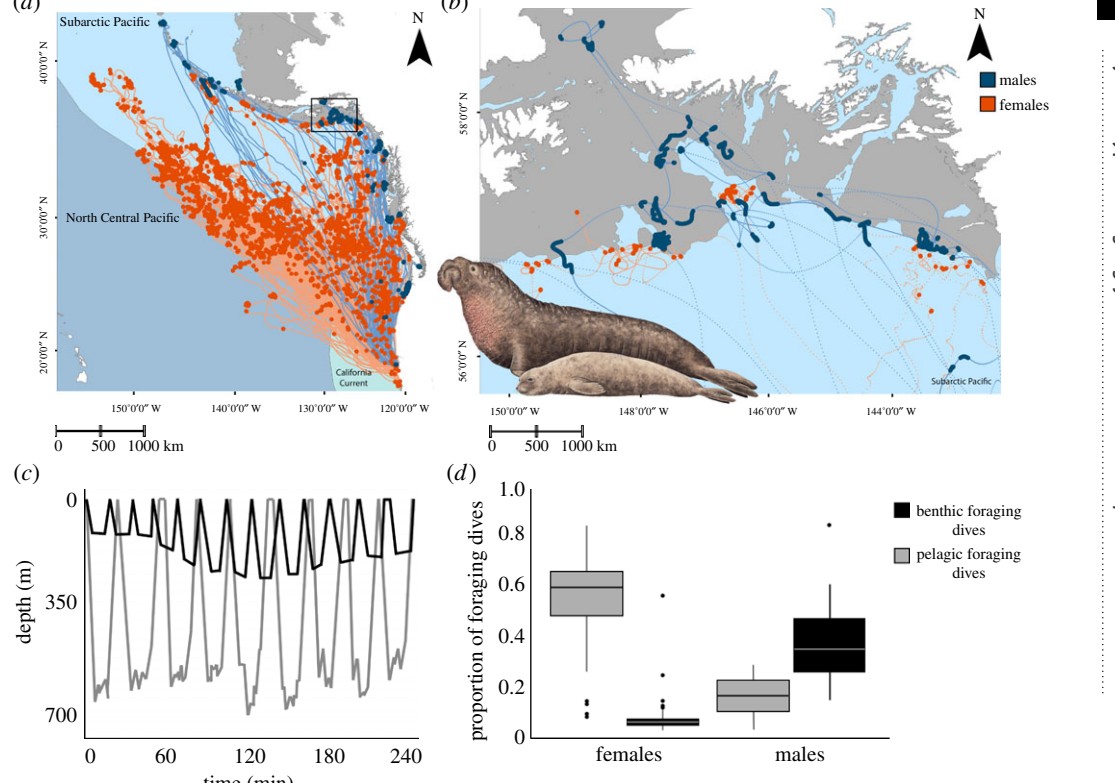

**Figure 1.** Comparison of satellite tracks and dive behaviour of 39 male and 178 female northern elephant seals (*Mirounga angustirostris*). (*a*) Males travel (blue lines) to coastal areas and forage (blue circles) on the continental shelf (grey area). Females travel (orange lines) and forage (orange circles) throughout the North Pacific. The three mesopelagic ecosystems (boundaries defined by Sutton *et al*. [35] used by northern elephant seals are colour-coded and labelled, with the California Current ecoregion in aqua, the Subarctic Pacific in light blue and the North Central Pacific in blue-grey. (*b*) Expanded view of differential male and female habitat use on/near the continental shelf in the Subarctic Pacific. Northern elephant seal illustration by Pieter Folkens. (*c*) Representative dive profile of daytime benthic foraging (black line) and pelagic foraging dives (grey line) from a male and female seal, respectively. The benthic foraging dives represented here occurred on the continental shelf, and the pelagic foraging dives were adjacent to the shelf edge. (*d*) Boxplots comparing the proportions of the two foraging dive types used by both sexes on their foraging trips: benthic foraging dives (black) and pelagic foraging dives (grey). Horizontal bars denote the 25th, 50th (median) and 75th quartile.

## 2. Results

### 2.1. Foraging strategies

Northern elephant seals used sex-specific foraging strategies, differing in their movement patterns, foraging locations and dive behaviour (figure 1). Hierarchical clustering analysis (HCA) of 130 seals (14 males, 116 females) and 31 foraging variables resulted in three foraging groups—one male strategy and two female strategies (post-breeding and post-moult; electronic supplementary material, tables S1–S3). Analyses of variance (ANOVAs) identified significant differences in the movement patterns and dive behaviour associated with each foraging strategy (table 1).

The male strategy was characterized by feeding on or near the continental shelf (within $33.50 \pm 68.86$ km of the shelf edge), as determined from the combination of a reduced transit speed (less than $2 \text{ km h}^{-1}$) and foraging dive behaviour (i.e. undertaking benthic or pelagic foraging dives; [36]) compared with pelagic-foraging females (post-breeding females: $478.09 \pm 353.63$ km, post-moult females: $621.17 \pm 344.23$ km; $\chi^2_{(2,149)} = 67.65$, $p < 0.001$). Most males (73%) took focused foraging trips (i.e. feeding events occurred at the farthest point of their trip from the breeding colony). Also, most males (84%) fed within the Subarctic Pacific ecoregion. Males had smaller foraging areas (mean ± standard deviation (s.d.); males: $32\,835 \pm 78\,860$ km²; post-breeding females: $56\,754 \pm 162\,759$ km²; post-moult females: $498\,344 \pm 587\,061$ km²; $\chi^2_{(2,149)} = 45.36$, $p < 0.001$) and undertook more benthic foraging

**Table 1.** Northern elephant seal movement and dive behaviour variables reported for males, post-breeding females and post-moult females. The three groups were determined from hierarchical clustering analyses of 31 variables of seal movement patterns and dive behaviour. Values are reported as mean ± s.d. Upper-case letters show significant differences between strategies from *post hoc* pairwise contrasts ($p \leq 0.05$).

| variable | males ($n = 32$) | post-breeding females ($n = 94$) | post-moult females ($n = 34$) |
|---|---|---|---|
| distance to continental shelf edge (km) | 33.50 ± 68.86[A] | 478.09 ± 353.63[B] | 621.17 ± 344.23[C] |
| proportion of time spent feeding | 0.53 ± 0.16 | 0.59 ± 0.14 | 0.54 ± 0.11 |
| foraging area (km$^2$) | 32,835 ± 78 860[A] | 56 754 ± 162 759[A] | 498 344 ± 587 061[B] |
| proportion of transit dives | 0.35 ± 0.13 | 0.30 ± 0.10 | 0.31 ± 0.09 |
| proportion of pelagic foraging dives (PFD) | 0.16 ± 0.08[A] | 0.55 ± 0.16[B] | 0.50 ± 0.13[B] |
| proportion of drift dives | 0.11 ± 0.05 | 0.12 ± 0.16 | 0.15 ± 0.12 |
| proportion of benthic foraging dives (BFD) | 0.40 ± 0.20[A] | 0.05 ± 0.07[B] | 0.05 ± 0.03[B] |
| max depth, day PFD (m) | 406.51 ± 83.43[A] | 593.51 ± 55.83[B,a] | 604.50 ± 46.08[B,a] |
| max depth, night PFD (m) | 364.59 ± 84.19[A] | 506.86 ± 37.70[B,a] | 483.78 ± 28.83[C,a] |
| dive duration, day PFD (min) | 23.90 ± 3.63[A] | 25.51 ± 2.83[A,a] | 27.01 ± 2.12[B,a] |
| dive duration, night PFD (min) | 22.75 ± 3.52[A] | 20.66 ± 2.21[B,a] | 22.57 ± 1.58[A,a] |
| bottom time, day PFD (min) | 12.50 ± 1.65 | 11.63 ± 2.25[a] | 12.21 ± 1.63[a] |
| bottom time, night PFD (min) | 11.97 ± 1.41[A] | 9.56 ± 1.65[B,a] | 11.05 ± 1.43[A,a] |
| post-dive interval, day PFD (min) | 2.59 ± 0.28[A] | 1.97 ± 0.29[B] | 2.17 ± 0.27[C] |
| post-dive interval, night PFD (min) | 2.50 ± 0.36[A] | 1.94 ± 0.31[B] | 2.31 ± 0.27[A] |
| no. vertical excursions, day PFD | 19.54 ± 2.33 | 18.39 ± 1.95[a] | 17.38 ± 2.02[a] |
| no. vertical excursions, night PFD | 19.12 ± 2.22[A] | 15.65 ± 2.84[B,a] | 16.09 ± 1.70[B,a] |
| efficiency, day PFD (unitless) | 0.47 ± 0.04[A] | 0.42 ± 0.04[B] | 0.42 ± 0.04[B,a] |
| efficiency, night PFD (unitless) | 0.48 ± 0.04[A] | 0.42 ± 0.04[B] | 0.44 ± 0.03[C,a] |
| max depth, day BFD (m) | 241.37 ± 104.52[A] | 250.26 ± 122.19[A,a] | 424.18 ± 119.52[B,a] |
| max depth, night BFD (m) | 219.59 ± 87.35 | 177.25 ± 125.88[A,a] | 239.63 ± 58.33[B,a] |
| dive duration, day BFD (min) | 20.72 ± 2.95[A] | 19.79 ± 4.2[A,a] | 27.72 ± 5.99[B,a] |
| dive duration, night BFD (min) | 19.92 ± 2.28[A] | 18.16 ± 3.15[A,a] | 23.04 ± 2.98[B,a] |
| bottom time, day BFD (min) | 13.49 ± 1.46[a] | 12.27 ± 2.88[A,a] | 14.82 ± 3.19[B,a] |
| bottom time, night BFD (min) | 12.06 ± 1.69[a] | 11.14 ± 2.40[A,a] | 13.10 ± 2.09[B,a] |
| post-dive interval, day BFD (min) | 2.43 ± 0.47[A] | 1.61 ± 0.43[B,a] | 2.15 ± 0.68[A] |
| post-dive interval, night BFD (min) | 3.43 ± 3.20 | 4.26 ± 10.16[a] | 4.48 ± 11.12 |
| no. vertical excursions, day BFD | 18.20 ± 2.65 | 22.19 ± 8.27[A] | 18.64 ± 6.39[B,a] |
| no. vertical excursions, night BFD | 16.18 ± 3.22 | 20.12 ± 8.08[A] | 15.05 ± 4.71[B,a] |
| efficiency, day BFD (unitless) | 0.59 ± 0.07[A] | 0.60 ± 0.10[A,a] | 0.52 ± 0.06[B] |
| efficiency, night BFD (unitless) | 0.54 ± 0.08 | 0.57 ± 0.09[A,a] | 0.53 ± 0.06[B] |

[a]Significant differences between day and night dive variables within a strategy ($p \leq 0.05$).

dives than females (males: $40 \pm 20\%$ of dives; post-breeding females: $5 \pm 7\%$ of dives; post-moult females: $5 \pm 3\%$ of dives; $\chi^2_{(2,149)} = 201.72$, $p < 0.001$). While males took two at-sea foraging trips each year, they did not use seasonally specific foraging strategies. The biannual male foraging trips were of equal duration (post-breeding: $124 \pm 21$ days, post-moult: $128 \pm 15$ days). Males foraged $53 \pm 16\%$ of their time at sea.

The female strategy was characterized by feeding in oceanic habitats greater than 450 km from the continental shelf edge. Females had larger core foraging areas and undertook more pelagic foraging dives than males (males: $16 \pm 8\%$ of dives; post-breeding females: $55 \pm 16\%$ of dives; post-moult females: $50 \pm 13\%$ of dives; $\chi^2_{(2,149)} = 84.85$, $p < 0.001$). Female dive behaviour showed diurnal

**Table 2.** Comparison of 2D and 3D foraging ranges (95% utilization distribution, UDs) and core foraging areas (50% UDs) and percentage of overlap of the foraging ranges and core foraging areas between male and female northern elephant seals.

| sex | kernel density | 2D | | 3D | |
|---|---|---|---|---|---|
| | | area (km$^2$) | % overlap | area (km$^3$) | % overlap |
| male | 95% | 188 | 9.92 | 51 509 | 21.6 |
| | 50% | 42 | 0 | 447 | 3.88 |
| female | 95% | 463 | 4.03 | 221 876 | 5.01 |
| | 50% | 93.3 | 0 | 278 | 6.25 |

patterns, as $t$-tests and Wilcoxon rank sum tests revealed significant differences between female daytime and night-time pelagic foraging dives. Daytime pelagic foraging dives were deeper (post-breeding trip: 593.51 ± 55.83 m, $t_{139.79}$ = 11.65, $p < 0.001$; post-moult trip: 604.50 ± 46.08 m, Wilcoxon rank sum test, $p < 0.001$) and longer (post-breeding trip: 25.51 ± 2.83 min, $t_{151.25}$ = 12.15, $p < 0.001$; post-moult trip: 27.01 ± 2.12 min, $t_{62.20}$ = 8.55, $p < 0.001$) than night-time dives (depth—post-breeding trip: 506.86 ± 37.70 m, post-moult trip: 483.78 ± 28.83 m; duration—post-breeding trip: 20.66 ± 2.21 min, post-moult trip: 22.57 ± 1.58 min). For post-breeding females, daytime dives also involved more bottom time (time spent at the bottom of a dive; 11.63 ± 2.25 min, $t_{146.62}$ = 6.64, $p < 0.001$) and had more vertical excursions (which represent prey capture attempts, [37]; 18.39 ± 1.95, $t_{151.11}$ = 5.30, $p < 0.001$) than night-time dives (bottom time—9.56 ± 1.65 min; vertical excursions—15.65 ± 2.84). The females' post-breeding foraging trip was shorter (76 ± 13 days) than the post-moult trip (220 ± 20 days). Post-breeding females foraged 59 ± 14% of their time at sea, and post-moult females foraged 54 ± 11%.

Females' post-breeding and post-moult trips were associated with different movement patterns and dive behaviour, resulting in seasonally specific foraging strategies. Most females on the post-breeding trip (76%) travelled to open ocean habitats—with post-breeding females staying closer to the continental shelf edge than post-moult females (478.09 ± 353.63 km and 621.17 ± 344 l.23 km from the shelf edge, respectively; $\chi^2_{(2,149)} = 67.65$, $p < 0.001$). Some post-breeding females undertook focused foraging trips (57%), while others (43%) foraged throughout the trip (i.e. feeding events occurred at many different points on the trip). Post-breeding females primarily fed in the Subarctic Pacific (46%) or North Central Pacific (32%). Post-breeding females had smaller foraging areas (56 754 ± 162 759 km$^2$) than post-moult females (498 344 ± 587 061 km$^2$; $\chi^2_{(2,149)} = 45.36$, $p < 0.001$). Post-breeding females also took shorter pelagic foraging dives (post-breeding females–day: 25.51 ± 2.83 min, night: 20.66 ± 2.21 min; post-moult females—day: 27.01 ± 2.12 min, night: 22.57 ± 1.58 min; day: $\chi^2_{(2,149)} = 14.56$, $p < 0.001$; night: $\chi^2_{(2,149)} = 131.18$, $p < 0.001$). Further, night-time pelagic foraging dives of post-breeding females were deeper (post-breeding females—506.86 ± 37.70 m, post-moult females: 483.78 ± 28.83 m; :$\chi^2_{(2,130)} = 131.18$, $p < 0.001$), had a shorter bottom time (post-breeding females: 9.56 ± 1.65 min, post-moult females: 11.05 ± 1.43 min; $\chi^2_{(2,130)} = 41.94$, $p < 0.001$), and lower dive efficiency (amount of time spent at the bottom of a dive relative to total dive duration; post-breeding females—0.42 ± 0.04; post-moult females—0.44 ± 0.03; $\chi^2_{(2,130)} = 34.67$, $p < 0.001$) than post-moult females. In comparison, most females on the post-moult trip travelled to open ocean habitats (65%) or coastal/open ocean habitats (29%). Most post-moult females (87%) fed throughout the trip. Post-moult females primarily fed in multiple ecoregions (58%), the Subarctic Pacific (23%) or the California Current (16%). Post-moult females had larger foraging areas, undertook longer pelagic foraging dives and night-time pelagic foraging dives were shallower, had longer bottom times, and higher dive efficiency than post-breeding females.

## 2.2. Overlap between strategies

The sexes showed little to no overlap in horizontal and vertical space use (table 2). When comparing two-dimensional (2D) satellite tracks, females only showed a 4% overlap with male foraging ranges (95% utilization distribution overlap index (UDOI) = 0.002). Females had no overlap with male core foraging areas (50% UDOI = 0.00). When comparing three-dimensional (3D) satellite tracks and vertical dives,

**Table 3.** Northern elephant seal foraging success variables (mean ± s.d.) for the three foraging strategies. Letters indicate significant differences between strategies based on *post hoc* pairwise contrasts ($p \leq 0.05$).

| Variable | males ($n = 32$) | post-breeding females ($n = 94$) | post-moult females ($n = 34$) |
|---|---|---|---|
| departure body mass (kg) | 1074.19 ± 194.29[A] | 334.72 ± 48.95[B] | 281.46 ± 36.34[C] |
| mass gain on trip (kg) | 458.44 ± 218.25[A] | 76.06 ± 31.22[B] | 232.82 ± 53.27[C] |
| mass gain rate on trip (kg d$^{-1}$) | 3.65 ± 1.61[A] | 0.99 ± 0.29[B] | 1.05 ± 0.21[B] |
| mass gain rate relative to feeding time (kg d$^{-1}$) | 5.66 ± 4.40[A] | 2.23 ± 3.08[B] | 2.14 ± 1.13[B] |
| proportion of mass gain on trip | 0.44 ± 0.23[A] | 0.23 ± 0.11[B] | 0.84 ± 0.21[C] |
| energy gain (MJ) | 8020 ± 2085[A] | 1439 ± 735[B] | 3747 ± 1009[C] |
| energy gain rate on trip (MJ d$^{-1}$) | 67.31 ± 23.10[A] | 18.38 ± 7.20[B] | 16.95 ± 3.96[B] |
| energy gain rate relative to feeding time (MJ d$^{-1}$) | 98.46 ± 66.71[A] | 32.81 ± 13.55[B] | 27.90 ± 11.34[B] |

females had a 5% overlap with male foraging ranges (95% UDOI = 0.04). Females had a 6.3% overlap with male core foraging areas (50% UDOI = 0.05).

## 2.3. Foraging success

Males had higher absolute mass and energy gain than females (table 3 and figure 2*a*,*b*). Specifically, males gained more mass (458 ± 218 kg) and energy (8020 ± 2085 MJ) when foraging at sea than females. Furthermore, relative to time spent feeding, males had increased relative rates of mass gain (6 ± 4 kg d$^{-1}$) and energy gain (99 ± 67 MJ d$^{-1}$) than females. Among females, post-breeding females gained less mass (96 ± 60 kg) and energy (1638 ± 918 MJ) than post-moult females (mass gain: 228 ± 67 kg, energy gain: 3794 ± 1052 MJ). Post-breeding and post-moult females, however, did not differ in their rates of mass and energy gain. Post-moult females, however, had the highest proportion of mass gain (the mean proportion of an individual's mass gained relative to their departure body mass) on the foraging trip (0.80 ± 0.26), compared with both males (0.44 ± 0.23) and post-breeding females (0.32 ± 0.25; $\chi^2_{(2,153)} = 405.44$, $p < 0.001$).

## 2.4. Mortality rate

Thirty-two per cent of the 217 satellite-tagged seals (39 males, 178 females) had their instruments stop transmitting at sea (males: $n = 25$, 64%; females: $n = 46$, 25%). Mechanical tag failure was differentiated from mortality events by analysing transmitted satellite position data and demographic/life-history data of instrumented seals. Mechanical tag failure accounted for 14% of satellite tags that stopped transmitting (males: $n = 8$, 20%; females: $n = 24$, 13%). The remaining 18% of tags stopped transmitting due to at-sea mortality (males: $n = 17$, 44%, females: $n = 22$, 12%; figure 2*c*–*e*).

Sex had a significant effect on mortality rate ($\chi^2_1 = 18.41$, $p < 0.001$). Males had a higher probability of mortality on foraging trips (0.45, 95% confidence interval (CI; 0.29, 0.61)) than females (0.12, 95% CI (0.08, 0.17)). Males that died were closer to the continental shelf edge (67 ± 120 km) than females at their last known location (470 ± 329 km; Wilcoxon rank sum test, $p < 0.001$). Most males (71%) died less than 30 km from the continental shelf edge, rather than in the open ocean (29%). Males died in the Subarctic Pacific (71%) or California Current (29%). In comparison, most females (91%) died in the open ocean, and only 9% died less than 30 km from the continental shelf edge. Females primarily died in the Subarctic Pacific (46%), followed by the California Current (32%) or North Central Pacific (23%).

# 3. Discussion

Northern elephant seals have a broad ecological niche. Males and females, however, occupy separate subsets of the species' overall niche space. Northern elephant seals use sex-specific foraging strategies

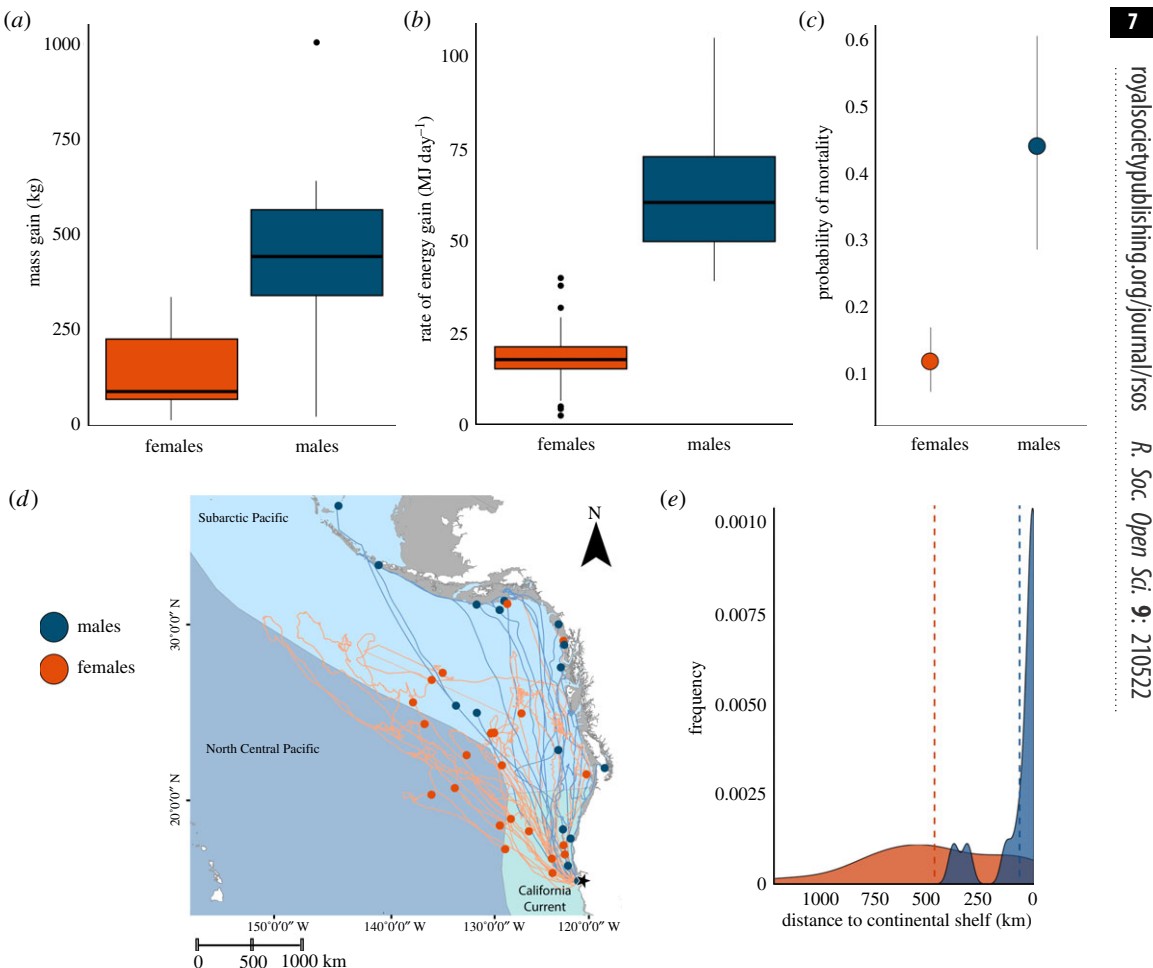

**Figure 2.** Comparison of foraging success and mortality rate metrics between male (blue) and female (orange) northern elephant seals. Boxplots compare sex-specific differences in (*a*) mass gain (kg), and (*b*) rate of energy gain (MJ d$^{-1}$) of 32 males and 128 females. Horizontal bars represent the 25th, 50th (median) and 75th quartile. (*c*) Probability of at-sea mortality for each sex on the foraging trip. Values are model mean ± 95% confidence intervals. (*d*) Satellite tracks of males (*n* = 17) and females (*n* = 22) that died at sea. Circles represent the point of last satellite transmission. The three mesopelagic ecosystems (boundaries defined by Sutton *et al.* [35]) used by northern elephant seals are colour-coded and labelled, with the California Current ecoregion in aqua, the Subarctic Pacific in light blue, and the North Central Pacific in blue-grey. (*e*) Density plot of dead males and females showing their distance to the continental shelf edge at their last satellite transmission. Vertical lines are the mean value for each sex.

that are characterized by specific movement patterns and dive behaviour. Males travel to coastal habitats along the eastern North Pacific coastline and feed continuously in small, localized foraging areas over the continental shelf. Males also take relatively shallow benthic foraging dives (approx. 230 m) to catch prey on or near the seafloor (e.g. sharks, cephalopods, fish; [38,39]).

Conversely, females travel throughout vast swathes of the North Pacific Ocean and feed in open ocean habitats. Females have large foraging areas and often travel between multiple prey patches. Females use deep (approx. 500–600 m) pelagic foraging dives to consume mesopelagic prey (e.g. fishes, squids; [40]). Females' dives also have a strong diurnal pattern to track vertically migrating prey. Within the general female strategy, females show seasonal foraging patterns. Females on the post-breeding trip have smaller foraging areas, shorter foraging dives with fewer prey capture attempts, and lower dive efficiency than females on the post-moult trip. The two female strategies are probably related to the different trip lengths, seasonal changes in prey and the physiological demands of gestation experienced by post-moult females [40,41]. These foraging behaviours are largely consistent with previous work [27–29,31,40]. Southern elephant seals (*Mirounga leonina*), the sister taxa to northern elephant seals, also show similar sexual segregation in foraging behaviours [42–45].

Habitat—specifically, coastal or open ocean—leads to strong convergence in marine megafauna movement patterns [46–48]. The open ocean is a highly dynamic environment that contains patchily

distributed prey [46–48]. Consequently, pelagic predators often travel large daily distances that are interspersed with short periods of concentrated movements when animals feed in a prey patch [46,47]. Coastal habitats, by comparison, are stable with high prey concentrations [46–48]. Coastal predators often travel short daily distances and show high variability in vertical movements when feeding [29,46–49]. Previous work suggests that both northern and southern elephant seals exhibit convergence in movement patterns and dive behaviour based on physical habitat [29–32,46–49]. Here, 6% of females travel to coastal habitats, like males. Unlike males, however, these females rarely, if ever, forage on the continental shelf. Instead, these coastal females feed in waters immediately adjacent to the shelf edge and use deep pelagic foraging dives to consume prey. Similarly, some males travel through pelagic habitats on their way to remote continental shelf habitat. However, they rarely, if ever, slow down until they begin benthic foraging over the continental shelf. Males and females, therefore, exhibit sexual segregation in their space use, which results in intraspecific niche divergence.

Males, as the larger sex, need absolutely more energy than females to support and sustain their body masses. Males also expend 5–6 times more energy during the breeding season compared with females, with the largest and most dominant males using the most energy [50]. The males' huge energy deficit must be—and is—recovered by feeding in benthic continental shelf ecosystems. Males have higher foraging success than females, gaining up to six times as much mass and acquiring energy 4.1 times faster. Males' high foraging success, though, appears to come at the cost of an increased mortality rate.

Males are six times more likely to die on the at-sea foraging trips than females. It is difficult to determine the cause of mortality for wide-ranging animals that mostly die at sea [34]. As a result, the frequency of all potential causes of northern elephant seal at-sea mortality (e.g. disease, injury, marine debris, predation, ship strikes, starvation) is unknown [23,25,34,51,52]. Predation and starvation are thought to be the leading causes of mortality in adult northern elephant seals [23,25], as opposed to injuries sustained during intense breeding competitions [51]. Here, we hypothesize that males experience increased predation pressure because their foraging areas overlap closely with those of known predators that results in males' high mortality rate.

White sharks (*Carcharodon carcharias*) and killer whales (*Orcinus orca*)—the two primary predators of northern elephant seals—hunt in coastal habitats along the eastern North Pacific coast [53]. Two-thirds of male mortality events occur in coastal habitats compared with only 9% of female mortality events. Males are found in killer whale stomachs more frequently [54–56] and are documented with more white shark bites than females [57,58]. Other North Pacific marine mammals also experience high levels of white shark and killer whale predation in coastal habitats [59]. Conversely, we find no significant differences in body condition or foraging behaviour between seals that lived and died that would support the starvation or injury hypotheses. Therefore, predation may be the primary cause of sex-specific mortality in northern elephant seals, but this requires further investigation. Specifically, future research is needed to: (i) define and quantify different sources of at-sea mortality in northern elephant seals, and (ii) determine satellite transmitted space use variables associated with different sources of at-sea mortality.

Our findings suggest that foraging strategy, potentially through sexual segregation of foraging habitat, drives sex-specific differences in northern elephant seal foraging success and mortality rates. Open ocean habitats provide sufficient prey resources for females to meet their physiological requirements while also appearing to minimize the risk of mortality. In contrast, males must feed in benthic continental ecosystems with abundant prey resources to meet their energetic requirements; however, these areas also appear to be associated with a higher risk of mortality.

Intraspecific niche divergence in northern elephant seals supports their different life-history strategies for maximizing fitness. Males reach physical and sexual maturity later in life and have lower annual survivorship than females [20,24,60]. Males must invest in growth to quickly obtain and then maintain the large body sizes required to be competitive in physical combats for access to mating opportunities and for protracted time spent onshore fasting [50,61]. Only a fraction of males that survive to adulthood—the largest and most dominant—successfully reproduce [20,60,61]. The extreme disparity in male reproductive success counterbalances the high mortality rate. A large proportion of males can die without affecting the population dynamics [60]. By contrast, females have higher annual survivorship and reach sexual maturity quicker than males [24,60]. The most successful females have long lifespans in which they reproduce annually until their death [24,25]. Females can meet their energetic demands in the pelagic environment and avoid the increased mortality risk, which prolongs their reproductive years.

Many sexually dimorphic taxa exhibit niche divergence, but it is often restricted to specific temporal periods around breeding and rearing offspring [18,19,62]. Male and female northern elephant seals,

however, operate in different ecological niches most of the year, except during the synchronous annual breeding season. Southern elephant seals also exhibit sexual dimorphism and sex-specific foraging strategies [42–45], with potential trade-offs between foraging habitat use and mortality rate [34]. Niche divergence may therefore have evolved early in elephant seals. A few closely related seals (e.g. grey seals, leopard seals) feature some form of sexual dimorphism with accompanying sex-specific differences in diet and/or foraging behaviour [63,64]. Most seals, however, do not [65,66]. Among pinnipeds (seals, sea lions and walruses) more broadly, some sexually dimorphic taxa show intrasexual differences in foraging patterns [63,67], while others do not [68,69]. It is therefore unlikely that sexual dimorphism and accompanying niche divergence represent the ancestral condition in pinnipeds. Instead, elephant seals provide a compelling example of how evolutionary and ecological processes can operate differently between the sexes.

Northern elephant seals exemplify many of the predictions associated with the evolution of sexual dimorphism. The consequences of niche divergence on fitness have remained elusive for many species [2,6–8]. Here, we show that northern elephant seal foraging strategies represent a trade-off between mortality risk and foraging reward. These risk-reward foraging patterns of northern elephant seals match predictions associated with the predation-risk hypotheses, which have mostly been examined in ungulates with mixed results [70,71]. The male northern elephant seal strategy requires high foraging success to sustain a large body size required for reproductive success; however, this comes at the cost of low survival. Alternatively, the female strategy promotes fitness by maximizing lifespan to increase reproductive success. Our findings highlight the interplay between sexual dimorphism and sex-specific behavioural and life-history strategies. Further, we demonstrate that trade-offs between mortality risk and foraging reward probably affect fitness by driving differences in the sexes' mortality rates. Finally, these results emphasize the importance of studying the biology of both sexes, especially for sexually dimorphic species where the sexes occupy completely different subsets of the overall species' niche space.

# 4. Material and methods

## 4.1. Instrumentation and animal handling

Bio-logging instruments were deployed on adult male ($n = 39$) and female ($n = 178$) seals at Año Nuevo State Park (San Mateo County, CA, USA) between 2006 and 2015 at the start of the post-breeding trip (females: February–May; males: March–August) and post-moult trip (females: May–January, males: August–January). Seals were instrumented with 0.5 W ARGOS satellite transmitters (SPOT or SPLASH tags, Wildlife Computers or Conductivity-Temperature-Depth tags, Sea Mammal Research Unit), time-depth recorders (TDRs; MK9 or MK10-AF, Wildlife Computers) and VHF radio transmitters (Advanced Telemetry Systems) using quick-set epoxy, high-tension mesh netting and cable ties. We chemically immobilized seals using an established protocol (detailed description in [31]) to attach the instruments and collect morphometric data and tissue samples [28,31,72,73]. All female and some male seals were previously flipper tagged as part of ongoing demographic studies of northern elephant seals at Año Nuevo. For flipper-tagged seals, we had long-term individual life-history and resighting data [25]. For untagged seals, a flipper tag was inserted into the webbing of each hind flipper during sedation. The flipper tags aided in individual identification during the biannual haul-outs. All instrumented seals also had their flipper tag number written on their sides in hair dye to facilitate visual identification upon their return to the colony. Upon the seal's return to the colony at the end of the foraging trip, seals were chemically immobilized to recover the instruments and to collect morphometric data and tissue samples.

## 4.2. Body composition

Each seal's body composition was measured on instrument deployment and recovery [28,30,31,73]. Length and girth measurements were taken at eight locations along the seal's body. Blubber thickness was measured with an ultrasound or a backfat meter at 12–18 locations along the body. We measured females' mass with a Dyna-Link digital scale attached to a tripod. Males' mass was estimated from the combination of lengths, girths and ultrasound measurements [50]. Mass was corrected for the time spent on shore. For females, mass change on shore was estimated using an equation derived from serial mass measurements of fasting seals from previous studies: mass change (kg d$^{-1}$)   0.51 +

0.0076 × mass, $n = 27$, $r^2 = 0.79$, $p < 0.01$ [73]. When females arrived after the post-moult trip, the recovery procedure occurred after parturition, and her pup's mass was added to the female's mass. For males, mass change on shore was estimated using a metabolic rate of two times the predicted metabolic rate for a mammal of equal size during the moult and 3.1 times during the breeding season [50,74,75] and fat and protein contributions to metabolism from Crocker et al. [50]. Energy gain was estimated assuming the adipose tissue was 90% lipid and lean tissue was 27% protein, with a gross energy content when mobilized of 37.3 kJ g$^{-1}$ for lipids and 23.5 kJ g$^{-1}$ for protein [76,77]. These estimates of body condition have been validated with those from the dilution of isotopically labelled water [77].

## 4.3. Data analysis

Statistical analyses and data analyses were conducted in R v. 3.3.3 [78]. Tracking and dive data were processed using standard filtering techniques and protocols for processing the satellite transmitter and TDR data as described in detail in [30,31]. Specifically, we truncated raw ARGOS tracks to the departure and arrival times from the breeding colony as identified from the diving record or sightings database. A speed, distance and angle filter removed unlikely position estimates (thresholds: 12 km h$^{-1}$ and 160°; argosfilter; [79]). The filter also examined secondary position calculations reported by ARGOS and replaced erroneous prior positions if speed/angle criteria were met [30,31]. Tracks were smoothed using a state-space model that yielded hourly estimates of position and incorporates estimates of at-sea ARGOS error (crawl; [80,81]). For females tracked in multiple years, we randomly removed repeat tracks so each seal was included in the analysis only once. All dives were georeferenced using the satellite track. Dive data were collected at sampling intervals between 1 and 8 s but were subsampled to 8 s for comparison. Dive behaviour was only analysed from recovered TDRs that recorded a full time series of data (Wildlife Computers MK9 or MK10-AF). The raw time series of depth measurements were analysed using a custom-written dive type script in Matlab (IKNOS Toolbox, Y Tremblay 2008, unpublished data; [30,31]). Identified dives were only retained if they exceeded 32 s in duration and 15 m in depth. Dives were classified into one of four types (pelagic foraging dives, benthic foraging dives, drift dives or transit dives) using a custom hierarchical classification program; this program was designed to detect the unique characteristics of each dive type (e.g. depth, shape, duration) as recorded by the TDR [30,31,36,72]. Females exhibit diel diving patterns, so we assigned each dive to day or night based on the solar zenith angle associated with each dive.

## 4.4. Foraging variables

We calculated the distance to the continental shelf edge (km) for each foraging location, which was defined by the 200 m depth isobath. We determined the proportion of time each seal spent feeding. We also calculated each seal's foraging region (km$^2$), defined as 95% contour area determined from the utilization distributions (UDs). The UDs were generated from kernel density analyses of 2D foraging locations (latitude + longitude) for each track using a 2 km cell size and default bandwidth in ArcGIS 10.3.1. Each track was assigned to a mesopelagic ecoregion ('ecoregion'; [35]) based on where the majority (greater than or equal to 50%) of foraging locations occurred. Each track was also assigned a habitat type [29,82]; tracks were categorized as coastal, coastal/open ocean or open ocean habitat. Lastly, each track was classified as 'focused' or 'throughout' based on the location of foraging points in relation to the furthest point from the breeding colony. 'Focused' trips occurred where feeding occurred at the most distant part of the track from the colony and less than five foraging locations were identified in other portions of the track. Trips were classified as 'throughout' when more than five foraging locations occurred outside the farthest point from the colony [83,84].

For each TDR record, we determined the proportion of the dive record in which seals used transit, pelagic foraging, drift and benthic foraging dives. For pelagic and benthic foraging dives, we calculated several dive metrics. We determined the mean maximum depth (m), bottom time (min) and the post-dive surface interval (min). We also calculated the mean number of vertical excursions at the bottom of each dive, which represent prey capture attempts [37] and the mean dive efficiency (bottom time/dive duration). Values closer to 0 indicate lower dive efficiency and values closer to 1 indicate higher dive efficiency [23,72]. We examined data for deviations from normality using density and Q-Q plots and a Shapiro–Wilks test. We then assessed variance with F-tests. Welch two-sample t-tests were used to compare diurnal patterns in seal dive behaviour. When data were not normally distributed, we ran non-

parametric Mann–Whitney–Wilcoxon tests. We applied a Bonferroni correction to account for the multiple comparisons, and adjusted $p$-values were used to assess significance.

## 4.5. Foraging success

We measured absolute and relative metrics of foraging success in northern elephant seals. We measured the seal's body mass at departure (kg) and determined the seal's total mass gain over the foraging trip (kg). We also determined the seal's rate of mass gain over the entire trip (kg d$^{-1}$) and the rate of mass gain relative to feeding time (kg d$^{-1}$). Feeding events were identified from satellite tracking and dive behaviour data. First, we calculated transit rate from the interpolated satellite tracks and then filtered the data for transit speeds less than 2 km h$^{-1}$, which are associated with foraging dive behaviour [36]. We further filtered the data to only include foraging locations that were associated with a foraging dive type (either pelagic and/or benthic foraging dives; [30,36]). Finally, we then calculated the proportion of time each seal spent feeding relative to their total time at sea. We calculated the seal's total energy gain on a trip (MJ), their total rate of energy gain (MJ d$^{-1}$), and the rate of energy gain relative to feeding time (MJ d$^{-1}$) based on the seal's rates of mass gain and body composition. We also calculated the proportion of a seal's mass gain on the foraging trip in relation to its departure mass to account for the sex-specific differences in body size.

## 4.6. Mortality rate

For each seal with an instrument that stopped transmitting (71 out of 217 seals), we did the following: (i) determined whether the seal was seen alive in subsequent years using historical records from 2006 to present from Año Nuevo State Park; and (ii) examined satellite position quality across the duration of the trip to ascertain whether the tag malfunctioned (e.g. degraded location quality signal over time) or functioned properly (e.g. random distribution of location qualities throughout the trip). Seals were resighted daily at Año Nuevo during the breeding and moult haul-outs. Seals were resighted weekly the rest of each year. We collected data on age, behaviour and haul-out history using archived resight records from the 1980s to present for each seal. Ground surveys and/or aerial photographic surveys (plane, drone) are conducted at least once each year at each northern elephant seal breeding colony in the USA, providing annual census data [33]. Double flipper tagging and retagging individuals helped account for biases that could result from flipper tag loss and/or inability to read tags [25]. To reduce bias due to instrumented seals returning to other colonies, we receive regular reports from stranding networks and colleagues observing seals at other colonies whenever seals flipper tagged and/or instrumented at Año Nuevo emigrate to other colonies [25,60]. Previous studies of female seals also show that emigration/immigration most frequently occurs in juveniles, with 90–95% of adult northern elephant seals showing high site fidelity to their natal colony [60]. If the tag was working correctly at the last transmission time and that seal was never seen again, the seal was then presumed dead. For each seal that died, we calculated the following metrics for the last known location: latitude and longitude, transit rate (m s$^{-1}$), distance to the continental shelf edge (km), ecoregion and portion of the trip (outward, farthest point or return). We used non-parametric Mann–Whitney–Wilcoxon tests to examine differences in the movement patterns between males and females that died. Adjusted $p$-values from a Bonferroni corrected were used to assess significance.

## 4.7. Statistical analyses

We conducted principal components analysis (PCA) on 31 foraging variables, encompassing seal movement patterns and dive behaviour (base-R, FactoMineR, missMDA, RColorBrewer, scales, tidyverse, gridExtra, here; [78,85–93]). Seals missing the majority of data were excluded from the analysis, resulting in a dataset of 14 males and 116 females. Variables were centred and scaled prior to PCA. A scree plot was used to examine natural breaking points in the variance. Principal components (PCs) with eigenvalues greater than or equal to 1.0 and that explained greater than or equal to 10.0% of the variation were retained. A coefficient correlation analysis was used to assess the contribution of each variable to each PC axis. Three PCs explained 56.2% of the total variation, and all variables were significantly correlated with one or more PC axes (electronic supplementary material, tables S1 and S2). Principal components 1–3 were analysed using HCA to examine naturally occurring clusters of individuals in the dataset. We created a dissimilarity matrix based on Euclidean distances and performed an agglomerative HCA using 'hclust' and Ward's linking method on the PC scores

(*tidyverse*, *cluster*, *factoextra*, *dendextend*; [93–96]). Elbow and average silhouette methods were used to determine the optimal number of clusters and assign each seal to a cluster (electronic supplementary material, table S3).

We set up generalized linear models for each foraging variable to determine foraging variables that best discriminated among clusters. Generalized linear models had a Gaussian distribution and an identity-link function. Cluster was the predictor variable. Each candidate model was compared with a null model (intercept only) using likelihood ratio tests of the null and residual deviances. Models were ranked using the Akaike information criterion corrected for small sample sizes (AICc; *AICcmodavg*; [97]). We evaluated the fit of the best model using an ANOVA (Type II, Wald's test, *car*; [98]). We then used estimated marginal means to perform *post hoc* pairwise contrasts between each cluster and used Tukey's method for adjusting the *p*-value for multiple comparisons (*emmeans*; [99]).

We examined the overlap between male and female 2D and 3D foraging ranges and core foraging areas by comparing UDs; this same approach was recently used in a study of nine individual southern elephant seals [100]. We used kernel density estimation to determine the 95% and 50% UDs for each sex, which represented foraging range (km$^2$) and core foraging areas (km$^2$), respectively [100]. The maximum dive depth was determined for each 2D foraging location (latitude + longitude) to create the 3D dataset. We only included locations and dives associated with foraging to examine overlap in the foraging space. A data-based 'plug-in' bandwidth selector (Hpi) was calculated for each dataset, and 2D and 3D kernel density UDs were calculated for males and females (*ks*; *KernSmooth*; *MASS*; [101–105]). We also calculated the proportion of overlap in the area (km$^2$, 2D-UD) and volume (km$^3$, 3D-UD) between the sexes and calculated the UDOI, which provided a measure of space-sharing use where values close to 0 represented no overlap and 1 indicated complete overlap (*misc3d*; [106,107]).

We tested for differences in mortality rate between the sexes using logistic regression models with a binomial distribution and logit-link function; these models tested the distribution of the data against a logistic curve to account for the categorial nature of the response variables (i.e. 'survived' = 0, 'died' = 1; [108,109]). Variables in the full model included sex, individual, tagging year and foraging trip (breeding or moult). We ran all possible combinations of the model that included sex as an explanatory variable using glm in R (electronic supplementary material, table S4). We compared the null and residual deviances of the models using likelihood ratio tests (*stats*, [78]). We ranked models using the Akaike information criterion and considered models of biological importance when ΔAIC was less than or equal to 7 [110]. We then tested the difference between our best-fitting model and the observed data using a Hosmer–Lemeshow goodness-of-fit test (*ResourceSelection*; [111]). We used an ANOVA to evaluate the fit of the best model and identified the most important explanatory variables with a Wald's test (*car*; [98]). We also modelled the probability of dying by sex across the foraging trips for our best-fitting model (*emmeans*; [99]).

Ethics. Behavioural research was approved by the Institutional Animal Care and Use Committee at the University of California, Santa Cruz and conducted under federal authorizations for marine mammal research under National Marine Fisheries Service permits 87-1743, 14636 and 19108.

Data accessibility. Life history, movement, dive behaviour, foraging success and mortality rate data for the northern elephant seals and the R scripts for data analysis and figure generation can be found on Dryad Digital Repository: https://doi.org/10.5061/dryad.c2fqz617f.

Authors' contributions. S.S.K.: conceptualization, formal analysis, funding acquisition, investigation, methodology, project administration, resources, visualization, writing—original draft, writing—review and editing; A.S.F.: formal analysis, resources, visualization, writing—review and editing; D.E.C.: formal analysis, methodology, resources, writing—review and editing; R.S.M.: conceptualization, resources, writing—review and editing; D.P.C.: conceptualization, data curation, funding acquisition, methodology, project administration, resources, writing—review and editing.

All authors gave final approval for publication and agreed to be held accountable for the work performed therein.

Competing interests. At the time of writing, Dr Ari Friedlaender is a Board Member of Royal Society Open Science, but had no involvement in the review or assessment of the paper.

Funding. This research was conducted as part of the Tagging of Pacific Predators program and supported in part by the National Ocean Partnership Program (N00014-02-1-1012); the Office of Naval Research (N00014-00-1-0880, N00014-03-1-0651, N00014-08-1-1195), the E&P Sound and Marine Life Joint Industry Project of the International Association of Oil and Gas Producers (JIP2207-23); the University of California Natural Reserve System, the NOAA Dr. Nancy Foster Scholarship Program, and the Moore, Packard and Sloan Foundations.

Acknowledgements. Volunteers, technicians, undergraduates and graduate students were instrumental in collecting and processing the project's data. We acknowledge the rangers and docents at Año Nuevo State Reserve and P. Morris for help resighting animals. We thank R. Condit for the development and maintenance of the demographic database. We thank C. Champagne, J. Estes, C. Goetsch, R. Holser, L. Hückstädt, S. C. Kienle, E. McHuron, S. Peterson and P. Robinson for suggestions that significantly contributed to the manuscript. Portions of the paper were developed from the dissertation of S.S. Kienle.

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
