## [Peer Review File · Royal Society Open Science]

Review History

RSOS-210522.R0 (Original submission)

Review form: Reviewer 1

Is the manuscript scientifically sound in its present form?

No

Are the interpretations and conclusions justified by the results?

Yes

Is the language acceptable?

Yes

Do you have any ethical concerns with this paper?

No

Have you any concerns about statistical analyses in this paper?

No

Recommendation?

Major revision is needed (please make suggestions in comments)

Comments to the Author(s)

I enjoyed reading this paper and am of the opinion that the data analyses are appropriate and overall conclusions interesting and defensible. Having said that, there are a number of things that require some clarification/editing in my opinion. I have commented directly on the manuscript (see Appendix A) with multiple queries/suggestions in the text and on the Tables and Figures, but the most important suggestions from my side are:

- No explanation is given for the dataset being split between foraging migrations for females, but not for males. My assumption is that this is simple a case of the sample size for males not allowing this – but this needs clarification;
- Why are the HCA results not presented? Yet referred to?
- Many of the results are reported in an unclear fashion – for example, some are given as means \pm standard deviations, while others appear to be reported as ranges, although these values are too close together to really represent ranges.
- The way the authors determined mortality is very much reliant on the survey effort that went into resighting animals returning to the colony and this is not explained. What sort of censusing effort went into resights? How (other than satellite tags) were the marked? What are the chances that animals were missed?
- There are some oversights of literature, particularly pertaining to literature on sex-specific foraging strategies in southern elephant seals, also within the context of predation risk. Most important of thesis probably Hindell et al. (2021), who's conclusions complement, if not mirror those of this paper. Also have a look at McIntyre et al (2010).

Hindell, M. A., McMahon, C. R., Jonsen, I., Harcourt, R., Arce, F., & Guinet, C. (2021). Inter- and intrasex habitat partitioning in the highly dimorphic southern elephant seal. *Ecology and Evolution*, 11(4), 1620–1633. <https://doi.org/10.1002/ece3.7147>

McIntyre, T., Tosh, C. A., Plötz, J., Bornemann, H., & Bester, M. N. (2010). Segregation in a sexually dimorphic mammal: a mixed-effects modelling analysis of diving behaviour in southern elephant seals. *Marine Ecology Progress Series*, 412, 293–304. <https://doi.org/10.3354/meps08680>

Review form: Reviewer 2**Is the manuscript scientifically sound in its present form?**

No

Are the interpretations and conclusions justified by the results?

No

Is the language acceptable?

Yes

Do you have any ethical concerns with this paper?

No

Have you any concerns about statistical analyses in this paper?

Yes

Recommendation?

Major revision is needed (please make suggestions in comments)

Comments to the Author(s)

RSOS-210522

General comments

This paper uses an impressive long-term data set from northern elephant seals to compare foraging strategies (geographic region targeted, plus diving behaviour) and survival in female and male seals. This is a valuable contribution making use of this rare, long term dataset. Many analyses have been performed and as such, the methods and results sections are dense, making them quite hard to follow. That said, there is so much to explain that I felt like some important methodological details were glossed over. It's really important to be as clear as possible about all the details of decisions and methodology that went into producing your results. These are important for convincing the reader of the validity and robustness of your results, and are central to reproducibility. If you don't have space to add details to the text, please consider adding it as a supplementary file.

Staying with the issue of reproducibility, I was not able to run any of the analyses due to lack of detail in the R code provided. Please add informative file names to the code that clearly link the script to the data provided. Please also add more explanation to the scripts to help a user follow your rationale and methodology. Lastly, please show that you have done model checking for all modelling approaches, and accounted for the many t-tests that are carried out by applying a Bonferroni correction to the value used to identify statistically significant correlations. $P=0.05$ should not be used when so many tests are carried out.

Specific comments

With respect to foraging success, it would help to give more details on what aspect of the Robinson et al 2010 paper you followed. Did you use all dive variables in their Table 1? Please specify what aspects of dive behaviour you used.

There are places in the manuscript where you assume quite a lot of reader knowledge about elephant seal diving ecology, for example on line 156 where you mention vertical excursions. What is the significance of vertical excursions for foraging success? The layout of the manuscript is such that the results come first, but I think it's important to provide the reader with enough contextual information to understand the results that are being presented. Again on line 159, you say that females fed throughout the trip. What was the evidence base you used to infer feeding?

Line 271: You say here that females minimise risk by foraging in the open ocean. Even with this impressive dataset, I don't think you are able to say that we know whether they are minimising overall risk. What your analyses show, is that females have a reduced mortality rate compared to males, and that this is associated with their pelagic foraging regions and diurnal diving behaviour. However, there are many aspects of "risk" that you are not able to quantify, such as competition from other females, competition from other species, environmental variability in resources, bias in the spatial origin of predators whose stomach contents has been analysed etc.

Line 320-323: How did you account for the differences in the data collection routines of different tagging devices? What are the caveats, if any, of having used all of these different devices? The differences appear both in the location technology and location error, as well as the way dive data were collected. Did you standardise all dive data to be the product of a broken-stick algorithm before carrying out dive analyses? Please provide more information here.

Line 218-219: You say here that females have shorter dives with fewer prey capture attempts, but I couldn't find any other mention of capture attempts or how they were defined.

Line 235: You say that males travel fast to the continental shelf and don't slow down to forage while transiting. How can you be confident that opportunistic feeding doesn't occur during transit? The dive profiles, if they were from a broken-stick algorithm, are very coarse-scale and mask detailed three-dimensional behaviour.

Line 352: What software package did you use to carry out the HCA analysis? How was dive shape determined? Please cite all software packages to give due credit. You cite three papers with respect to dive variable extraction, but it would be useful to also mention what method or algorithm you used to determine and define shape.

Though this dataset is the largest on northern elephant seals, the numbers of seals that died is relatively small. Looking at Figure 2D, I wouldn't say there is crystal clear separation between the regions where males and females died, so I would be a little bit more cautious about reporting the conclusions of the survival analysis, which may change as the dataset on mortality grows.

Data and code accessibility

I managed to download all of the scripts and the .xlsx file but I wasn't able to run the analyses because it wasn't clear which data needed to be used in each script. All of the scripts refer to "file name.csv" as the data, but there is no file with this name. There are different sheets in the .xlsx but it's not clear which one of them should be used where. At the moment it's not possible to reproduce the analyses. Please add informative file names to the scripts. I am not able to comment on many of the scripts without having access to the data.

Script Mortality_Analyses:

Here you fit a binomial GLM to the data on survival, I think from the All Animals sheet. You do an ANOVA and a comparison of group means by sex, but you don't seem to do any model checking. Were the assumptions of this model met? How much variability in the data did the model explain? Without information on model fit and diagnostics, the reader doesn't have information about whether they can trust the results.

Script t-tests:

In the t-tests script you carry out 119 separate t-tests. As the number of t-tests increases, so does the chance of finding spurious correlations between groups. The Bonferroni correction can be used to compensate for this. Did you apply the Bonferroni correction in interpreting the results of your t-tests? I also wondered if you checked that the two groups going into each of your t-tests had equal variances? That's an assumption of the t-tests you've used. You have quite big sample sizes it might be fine, but I couldn't tell without reading in the data. If you didn't apply a Bonferroni correction, please do so and revise the p-values used to help identify statistically significant results.

Script Linear models ANOVA (both scripts):

Similarly to the survival analysis script, there is no model checking in this script. I would strongly recommend doing model checking for each of the models that you fit.

Decision letter (RSOS-210522.R0)

Dear Dr Kienle

The Editors assigned to your paper RSOS-210522 "Trade-offs between foraging reward and mortality risk drive sex-specific foraging strategies in a sexually dimorphic mammal" have now received comments from reviewers and would like you to revise the paper in accordance with the reviewer comments and any comments from the Editors. Please note this decision does not guarantee eventual acceptance.

Please submit your revised manuscript and required files (see below) no later than 21 days from today's (ie 02-Aug-2021) date. Note: the ScholarOne system will 'lock' if submission of the revision is attempted 21 or more days after the deadline. If you do not think you will be able to meet this deadline please contact the editorial office immediately.

on behalf of Dr Denise Greig (Associate Editor) and Kevin Padian (Subject Editor)
openscience@royalsociety.org

Subject Editor Comments to Author:

Thanks for your submission and I hope you find the comments useful. In addition, I think it would be beneficial to change your title from "mammal" to "the Northern elephant seal,

Mirounga angustirostris." This helps readers who see titles to identify their interest, and helps in indexing. You are only studying one species and so it makes sense to identify it. Thanks.

Associate Editor Comments to Author (Dr Denise Greig):

Thank you for submitting this interesting analysis of an impressive long term dataset. I especially enjoyed the data rich figures.

Finding reviewers took some time, however, both provided constructive comments for clarifying the methods, tightening up the references, and improving the consistency of the results.

I agree with the comment from reviewer 2 that while you convincingly show that females have lower mortality than males and that this is associated with pelagic foraging regions and diurnal dive behavior, these findings do not necessarily translate to minimizing risk by foraging in the open ocean as there are other risk factors that cannot be addressed with these data.

I hope you will be willing to address the reviewer comments and I look forward to reading the next version.

Reviewer comments to Author:

Reviewer: 1

Comments to the Author(s)

I enjoyed reading this paper and am of the opinion that the data analyses are appropriate and overall conclusions interesting and defensible. Having said that, there are a number of things that require some clarification/editing in my opinion. I have commented directly on the manuscript with multiple queries/suggestions in the text and on the Tables and Figures (please see attachment: "RSOS-210522_reviewer - Copy"), but the most important suggestions from my side are:

- No explanation is given for the dataset being split between foraging migrations for females, but not for males. My assumption is that this is simple a case of the sample size for males not allowing this – but this needs clarification;
- Why are the HCA results not presented? Yet referred to?
- Many of the results are reported in an unclear fashion – for example, some are given as means \pm standard deviations, while others appear to be reported as ranges, although these values are too close together to really represent ranges.
- The way the authors determined mortality is very much reliant on the survey effort that went into resighting animals returning to the colony and this is not explained. What sort of censusing effort went into resights? How (other than satellite tags) were the marked? What are the chances that animals were missed?
- There are some oversights of literature, particularly pertaining to literature on sex-specific foraging strategies in southern elephant seals, also within the context of predation risk. Most important of thesis probably Hindell et al. (2021), who's conclusions complement, if not mirror those of this paper. Also have a look at McIntyre et al (2010).

Hindell, M. A., McMahon, C. R., Jonsen, I., Harcourt, R., Arce, F., & Guinet, C. (2021). Inter- and intrasex habitat partitioning in the highly dimorphic southern elephant seal. *Ecology and Evolution*, 11(4), 1620–1633. <https://doi.org/10.1002/ece3.7147>

McIntyre, T., Tosh, C. A., Plötz, J., Bornemann, H., & Bester, M. N. (2010). Segregation in a sexually dimorphic mammal: a mixed-effects modelling analysis of diving behaviour in southern elephant seals. *Marine Ecology Progress Series*, 412, 293–304. <https://doi.org/10.3354/meps08680>

Reviewer: 2
Comments to the Author(s)
RSOS-210522

General comments

This paper uses an impressive long-term data set from northern elephant seals to compare foraging strategies (geographic region targeted, plus diving behaviour) and survival in female and male seals. This is a valuable contribution making use of this rare, long term dataset. Many analyses have been performed and as such, the methods and results sections are dense, making them quite hard to follow. That said, there is so much to explain that I felt like some important methodological details were glossed over. It's really important to be as clear as possible about all the details of decisions and methodology that went into producing your results. These are important for convincing the reader of the validity and robustness of your results, and are central to reproducibility. If you don't have space to add details to the text, please consider adding it as a supplementary file.

Staying with the issue of reproducibility, I was not able to run any of the analyses due to lack of detail in the R code provided. Please add informative file names to the code that clearly link the script to the data provided. Please also add more explanation to the scripts to help a user follow your rationale and methodology. Lastly, please show that you have done model checking for all modelling approaches, and accounted for the many t-tests that are carried out by applying a Bonferroni correction to the value used to identify statistically significant correlations. $P=0.05$ should not be used when so many tests are carried out.

Specific comments

With respect to foraging success, it would help to give more details on what aspect of the Robinson et al 2010 paper you followed. Did you use all dive variables in their Table 1? Please specify what aspects of dive behaviour you used.

There are places in the manuscript where you assume quite a lot of reader knowledge about elephant seal diving ecology, for example on line 156 where you mention vertical excursions. What is the significance of vertical excursions for foraging success? The layout of the manuscript is such that the results come first, but I think it's important to provide the reader with enough contextual information to understand the results that are being presented. Again on line 159, you say that females fed throughout the trip. What was the evidence base you used to infer feeding?

Line 271: You say here that females minimise risk by foraging in the open ocean. Even with this impressive dataset, I don't think you are able to say that we know whether they are minimising overall risk. What your analyses show, is that females have a reduced mortality rate compared to males, and that this is associated with their pelagic foraging regions and diurnal diving behaviour. However, there are many aspects of "risk" that you are not able to quantify, such as competition from other females, competition from other species, environmental variability in resources, bias in the spatial origin of predators whose stomach contents has been analysed etc.

Line 320-323: How did you account for the differences in the data collection routines of different tagging devices? What are the caveats, if any, of having used all of these different devices? The differences appear both in the location technology and location error, as well as the way dive data were collected. Did you standardise all dive data to be the product of a broken-stick algorithm before carrying out dive analyses? Please provide more information here.

Line 218-219: You say here that females have shorter dives with fewer prey capture attempts, but I couldn't find any other mention of capture attempts or how they were defined.

Line 235: You say that males travel fast to the continental shelf and don't slow down to forage while transiting. How can you be confident that opportunistic feeding doesn't occur during transit? The dive profiles, if they were from a broken-stick algorithm, are very coarse-scale and mask detailed three-dimensional behaviour.

Line 352: What software package did you use to carry out the HCA analysis? How was dive shape determined? Please cite all software packages to give due credit. You cite three papers with respect to dive variable extraction, but it would be useful to also mention what method or algorithm you used to determine and define shape.

Though this dataset is the largest on northern elephant seals, the numbers of seals that died is relatively small. Looking at Figure 2D, I wouldn't say there is crystal clear separation between the regions where males and females died, so I would be a little bit more cautious about reporting the conclusions of the survival analysis, which may change as the dataset on mortality grows.

Data and code accessibility

I managed to download all of the scripts and the .xlsx file but I wasn't able to run the analyses because it wasn't clear which data needed to be used in each script. All of the scripts refer to "file name.csv" as the data, but there is no file with this name. There are different sheets in the .xlsx but it's not clear which one of them should be used where. At the moment it's not possible to reproduce the analyses. Please add informative file names to the scripts. I am not able to comment on many of the scripts without having access to the data.

Script Mortality_Analyses:

Here you fit a binomial GLM to the data on survival, I think from the All Animals sheet. You do an ANOVA and a comparison of group means by sex, but you don't seem to do any model checking. Were the assumptions of this model met? How much variability in the data did the model explain? Without information on model fit and diagnostics, the reader doesn't have information about whether they can trust the results.

Script t-tests:

In the t-tests script you carry out 119 separate t-tests. As the number of t-tests increases, so does the chance of finding spurious correlations between groups. The Bonferroni correction can be used to compensate for this. Did you apply the Bonferroni correction in interpreting the results of your t-tests? I also wondered if you checked that the two groups going into each of your t-tests had equal variances? That's an assumption of the t-tests you've used. You have quite big sample sizes it might be fine, but I couldn't tell without reading in the data. If you didn't apply a Bonferroni correction, please do so and revise the p-values used to help identify statistically significant results.

Script Linear models ANOVA (both scripts):

Similarly to the survival analysis script, there is no model checking in this script. I would strongly recommend doing model checking for each of the models that you fit.

===PREPARING YOUR MANUSCRIPT===

one version identifying all the changes that have been made (for instance, in coloured highlight, in bold text, or tracked changes);
 a 'clean' version of the new manuscript that incorporates the changes made, but does not highlight them. This version will be used for typesetting if your manuscript is accepted.

===PREPARING YOUR REVISION IN SCHOLARONE===

- Any electronic supplementary material (ESM).
- If you are requesting a discretionary waiver for the article processing charge, the waiver form must be included at this step.
- If you are providing image files for potential cover images, please upload these at this step, and inform the editorial office you have done so. You must hold the copyright to any image provided.
- A copy of your point-by-point response to referees and Editors. This will expedite the preparation of your proof.

- Ensure that your data access statement meets the requirements at <https://royalsociety.org/journals/authors/author-guidelines/#data>. You should ensure that you cite the dataset in your reference list. If you have deposited data etc in the Dryad repository, please include both the 'For publication' link and 'For review' link at this stage.
- If you are requesting an article processing charge waiver, you must select the relevant waiver option (if requesting a discretionary waiver, the form should have been uploaded at Step 3 'File upload' above).
- If you have uploaded ESM files, please ensure you follow the guidance at <https://royalsociety.org/journals/authors/author-guidelines/#supplementary-material> to include a suitable title and informative caption. An example of appropriate titling and captioning may be found at https://figshare.com/articles/Table_S2_from_Is_there_a_trade-off_between_peak_performance_and_performance_breadth_across_temperatures_for_aerobic_scope_in_teleost_fishes_/3843624.

Author's Response to Decision Letter for (RSOS-210522.R0)

See Appendix B.

RSOS-210522.R1 (Revision)

Review form: Reviewer 1

Is the manuscript scientifically sound in its present form?

Yes

Are the interpretations and conclusions justified by the results?

Yes

Is the language acceptable?

Yes

Do you have any ethical concerns with this paper?

No

Have you any concerns about statistical analyses in this paper?

No

Recommendation?

Accept as is

Comments to the Author(s)

Thank you very much for the clear and thorough responses to my comments on the previous version of your article. Other than one, minor suggestion (see below), I really do not have any further substantial concerns that require addressing.

- In the Methods/Foraging success section you mention that you only retained foraging locations associated with pelagic and benthic foraging dives. The wording choice here was a bit confusing to me initially as it seemed to suggest locations where seals were performing both pelagic and benthic dives (and not either of the two). Perhaps consider rephrasing slightly for clarity.

Decision letter (RSOS-210522.R1)

Dear Dr Kienle

On behalf of the Editors, we are pleased to inform you that your Manuscript RSOS-210522.R1 "Trade-offs between foraging reward and mortality risk drive sex-specific foraging strategies in sexually dimorphic northern elephant seals" has been accepted for publication in Royal Society Open Science subject to minor revision in accordance with the referees' reports. Please find the referees' comments along with any feedback from the Editors below my signature.

Please submit your revised manuscript and required files (see below) no later than 7 days from today's (ie 06-Dec-2021) date. Note: the ScholarOne system will 'lock' if submission of the revision is attempted 7 or more days after the deadline. If you do not think you will be able to meet this deadline please contact the editorial office immediately.

on behalf of Dr Denise Greig (Associate Editor) and Kevin Padian (Subject Editor)
 openscience@royalsociety.org

Associate Editor Comments to Author (Dr Denise Greig):

Associate Editor: 1

Comments to the Author:

Thank you for your really thorough response to the reviewers: this revision is really well done. There is one request from one of the reviewers and I just have a few minor comments: Regarding "Distance to continental shelf" - is this the distance to the edge of the continental shelf? And does it matter if the seal is beyond the shelf or between the shelf edge and the coast? I am having trouble visualizing what mean +/- sd means when one tail of the distribution might place the animal on land. I think how this is calculated should be clarified in the methods (Line 404), and also in the results at line 133 where it is unclear what is being measured and compared among the three diving strategies. [for example, "The male strategy was characterized by feeding on or near the continental shelf (within 33.5+/-68.86 km of the shelf edge)". If this is a correct interpretation.

Also, in Table 1, the variable name could be adjusted to "Distance to continental shelf edge" or whatever is most accurate.

Foraging Success - I am struggling a little bit with these metrics. To me, the one that makes the most sense is the proportion of mass gain. The post-molt females exhibited the greatest proportional mass gain which makes sense given active gestation.

However, based on the rest of the metrics, the males clearly need and acquire more absolute energy.

I think maybe a recognition in the methods or results that these metrics measure slightly different things would be helpful. And that the need for more absolute energy is what you think drives the sex difference in where they forage (if I have that right).

I do appreciate lines 242-244 in the discussion, where you acknowledge the demands of gestation.

Line 385, please change examines to examined

Line 386, please change replaces to replaced

Reviewer comments to Author:

Reviewer: 1

Comments to the Author(s)

Thank you very much for the clear and thorough responses to my comments on the previous version of your article. Other than one, minor suggestion (see below), I really do not have any further substantial concerns that require addressing.

- In the Methods/Foraging success section you mention that you only retained foraging locations associated with pelagic and benthic foraging dives. The wording choice here was a bit confusing to me initially as it seemed to suggest locations where seals were performing both pelagic and benthic dives (and not either of the two). Perhaps consider rephrasing slightly for clarity.

===PREPARING YOUR MANUSCRIPT===

one version should clearly identify all the changes that have been made (for instance, in coloured highlight, in bold text, or tracked changes);

===PREPARING YOUR REVISION IN SCHOLARONE===

-- Ensure that your data access statement meets the requirements at https://royalsociety.org/journals/authors/author-guidelines/#data.

You should ensure that you cite the dataset in your reference list. If you have deposited data etc in the Dryad repository, please only include the 'For publication' link at this stage. You should remove the 'For review' link.

-- If you are requesting an article processing charge waiver, you must select the relevant waiver option (if requesting a discretionary waiver, the form should have been uploaded, see 'File upload' above).

-- If you have uploaded any electronic supplementary (ESM) files, please ensure you follow the guidance at <https://royalsociety.org/journals/authors/author-guidelines/#supplementary-material> to include a suitable title and informative caption. An example of appropriate titling and captioning may be found at https://figshare.com/articles/Table_S2_from_Is_there_a_trade-off_between_peak_performance_and_performance_breadth_across_temperatures_for_aerobic_scope_in_teleost_fishes_/3843624.

Author's Response to Decision Letter for (RSOS-210522.R1)

See Appendix C.

Decision letter (RSOS-210522.R2)

Dear Dr Kienle,

I am pleased to inform you that your manuscript entitled "Trade-offs between foraging reward and mortality risk drive sex-specific foraging strategies in sexually dimorphic northern elephant seals" is now accepted for publication in Royal Society Open Science.

on behalf of Dr Denise Greig (Associate Editor) and Kevin Padian (Subject Editor)
openscience@royalsociety.org

Appendix A**ROYAL SOCIETY
OPEN SCIENCE****Trade-offs between foraging reward and mortality risk drive
sex-specific foraging strategies in a sexually dimorphic
mammal**

Journal:	Royal Society Open Science
Manuscript ID	RSOS-210522
Article Type:	Research
Date Submitted by the Author:	26-Mar-2021
Complete List of Authors:	Kienle, Sarah; Baylor University, Biology; University of California Santa Cruz, Ecology and Evolutionary Biology Friedlaender, Ari; University of California Santa Cruz Long Marine Laboratory, Institute for Marine Sciences Crocker, Daniel; Sonoma State University, Biology Mehta, Rita; University of California Santa Cruz, Ecology& Evolutionary Biology Costa, Daniel; University of California Santa Cruz, Ecology and Evolutionary Biology
Subject:	behaviour < BIOLOGY, ecology < BIOLOGY, physiology < BIOLOGY
Keywords:	Niche divergence, Feeding, Fitness, Marine mammal, Spatial ecology, Survival
Subject Category:	Organismal and Evolutionary Biology

Author-supplied statements

Relevant information will appear here if provided.

Ethics

Does your article include research that required ethical approval or permits?:

Yes

Statement (if applicable):

Behavioral research was approved by the Institutional Animal Care and Use Committee at the University of California, Santa Cruz and conducted under federal authorizations for marine mammal research under National Marine Fisheries Service permits 87-1743, 14636, and 19108.

Data

It is a condition of publication that data, code and materials supporting your paper are made publicly available. Does your paper present new data?:

Yes

Statement (if applicable):

Life history, movement, dive behavior, foraging success, and mortality rate data for the northern elephant seals and the R scripts for data analysis and figure generation can be found on Dryad <https://doi.org/10.5061/dryad.c2fqz617f>.

The link during the peer-review process is:

https://datadryad.org/stash/share/b7_99YEQgbyKptOXBn-6gSTuMBjQhznW-k53EBGCGr0.

Conflict of interest

I/We declare a competing interest

Statement (if applicable):

At the time of writing, Dr Ari Friedlaender is a Board Member of Royal Society Open Science, but had no involvement in the review or assessment of the paper.

Authors' contributions

This paper has multiple authors and our individual contributions were as below

Statement (if applicable):

Conceptualization: SSK, DPC, RSM; Methodology: SSK, DPC, DEC; Formal analysis: SSK, DEC, ASF; Resources: All; Writing – original draft: SSK; Writing – Revision/editing: All; Project administration: SSK, DPC; Funding acquisition: SSK, DPC

1
2

[revised manuscript text omitted]

(SD); $74,911\pm 203,029$ km²; $F_{2,138}=13.82$, $p<0.001$] and undertook more benthic foraging dives
($40\pm 20\%$ of dives; $F_{2,125}=76.27$, $p<0.001$) than females. Males took two at-sea foraging trips
each year. The biannual male foraging trips were of equal duration (post-breeding: 124 ± 21 days,
post-molt: 128 ± 15 days). Males foraged $53\pm 16\%$ of their time at sea.

The female strategy was characterized by feeding in oceanic habitats >500 km from the
continental shelf. Females had larger core foraging areas ($171,159-334,526$ km²) and undertook
more pelagic foraging dives ($54\pm 14\%$ of dives; $F_{2,125}=53.01$, $p<0.001$) than males. Female dive
behavior showed diurnal patterns. Further, t-tests revealed significant differences between female
daytime and nighttime pelagic foraging dives. Daytime pelagic foraging dives were deeper ($597-$
605 m; post-breeding trip: $t_{150,45}=13.61$, $p<0.001$; post-molt trip: $t_{47,3}=11.78$, $p<0.001$) and longer
($25-28$ min; post-breeding trip: $t_{48,5}=7.60$, $p<0.001$; post molt trip: $t_{162,3}=16.75$, $p<0.001$) than
nighttime dives (depth: $494-503$ m; duration: $20-24$ min). The females' post-breeding foraging
trip was shorter (76 ± 13 days) than the post-molt trip (220 ± 20 days). On both trips, females
foraged $58\pm 12\%$ of their time at sea.

Females' post-breeding and post-molt trips were associated with different movement
patterns and dive behavior, resulting in seasonally-specific foraging strategies. Most females on
the post-breeding trip (76%) traveled to open ocean habitats. Some post-breeding females
undertook focused foraging trips (57%), while others (43%) foraged throughout the trip. Post-
breeding females primarily fed in the Subarctic Pacific (46%) or North Central Pacific (32%).
Post-breeding females had smaller core foraging areas ($171,159\pm 413,562$ km²) than post-molt
females. Also, post-breeding females took shorter pelagic foraging dives ($20-25$ min;

$F_{2,125}=17.39$, $p<0.01$) that had fewer vertical excursions (15-18; $F_{2,125}=29$  $p<0.001$) and lower
 dive efficiency (0.41-0.42; $F_{2,125}=36.76$, $p<0.001$) than post-molt females. In comparison, most
 females on the post-molt trip traveled to open ocean habitats (65%) or coastal/open ocean
 habitats (29%). Most post-molt females (87%) fed throughout the trip. Post-molt females
 primarily fed in multiple ecoregions (58%), the Subarctic Pacific (23%), or the California
 Current (16%). Post-molt females had larger core foraging areas ($334,526\pm 464,054$ km²),
 undertook longer pelagic foraging dives (24-28 min) that had more vertical excursions (17.44-
 18.60) and higher dive efficiency (0.43-0.46) than post-breeding females.

*Overlap Between Strategies*

The sexes showed little to no overlap in horizontal and vertical space use (Table 2).
 When comparing two-dimensional (2D) satellite tracks, females only showed a 4% overlap with
 male foraging ranges [95% Utilization Distribution Overlap Index (UDOI)=0.002]. Females had
 no overlap with male core foraging areas (50% UDOI=0.00). When comparing three-
 dimensional (3D) satellite tracks and vertical dives, females had a 5% overlap with male
 foraging ranges (95% UDOI=0.04). Females had a 6.3% overlap with male core foraging areas
 (50% UDOI=0.05).

*Foraging Success*

Males had higher foraging success—increased relative rates of mass and energy gain—
 than females (Table 3, Fig. 2A-B). Males gained more mass (458 ± 218 kg; $F_{2,138}=71.68$, $p<0.001$)
 and energy ($8,020\pm 2,085$ MJ; $F_{2,101}=73.73$, $p<0.001$) when foraging at-sea than females.  178 Furthermore, relative to time spent feeding, males had faster rates of mass gain (6 ± 4 kg day⁻¹;
 $F_{2,121}=20.22$, $p<0.001$) and energy gain (99 ± 67 MJ day⁻¹; $F_{2,90}=13.65$, $p<0.001$) than females.

Post-breeding females gained less mass (96 ± 60 kg) and energy ($1,638\pm 918$ MJ) than
 post-molt females (mass gain: 228 ± 67 kg, energy gain: $3,794\pm 1,052$ MJ). Post-breeding and

[revised manuscript text omitted]

We ran linear models for each variable with cluster as the predictor variable to determine
variables that best discriminated among clusters. An ANOVA was used to determine significant
differences among clusters (car). We used least-square means to perform Tukey post-hoc
pairwise contrasts between each cluster (lsmeans; 82). We examined residual plots of all feeding
variables for obvious deviations from normality or homoscedasticity using histograms and Q-Q
plots. When deviations from normality were observed, we used log and square root
transformations so that all variables approached a normal distribution.

We examined the overlap between male and female 2D and 3D foraging ranges and core 42
foraging areas by comparing UD. We used kernel density estimation to determine the 95% and
50% UD for each sex, which represented foraging range (km^2) and core foraging areas (km^2),
respectively. The maximum dive depth was determined for each 2D foraging location (latitude +
longitude) to create the 3D dataset. We only included locations and dives associated with
foraging to examine overlap in the foraging space. A data-based "plug-in" bandwidth selector
(Hpi) was calculated for each dataset, and 2D and 3D kernel density UD were calculated for
males and females (ks; 83-84). We calculated the proportion of overlap in the area (km^2 , 2D-UD)
and volume (km^3 , 3D-UD) between the sexes and calculated the UDUI, which provided a
measure of space-sharing use where values close to 0 represented no overlap and 1 indicated
complete overlap (82, 85).

We tested for differences in mortality rate between the sexes. We fit a generalized linear
model with a binomial distribution and logit-link function with sex and foraging as fixed effects.
The significance of the fixed effects was determined using a χ^2 test (car). We also modeled the
probability of dying by sex across the foraging trips (emmeans).

ACKNOWLEDGEMENTS

Volunteers, technicians, undergraduates, and graduate students were instrumental in
collecting and processing the project's data. We acknowledge the rangers and docents at Año
Nuevo State Reserve and P Morris for help resighting animals. We thank R Condit for the
development and maintenance of the demographic database. We thank C Champagne, J Estes, C
Goetsch, R Holser, L Hückstädt, SC Kienle, E McHuron, S Peterson, and P Robinson for
suggestions that significantly contributed to the manuscript. Portions of the paper were
developed from the dissertation of SS Kienle. This research was conducted as part of the
Tagging of Pacific Predators program and supported in part by the National Ocean Partnership
Program (N00014-02-1-1012); the Office of Naval Research (N00014-00-1-0880, N00014-03-
1-0651, N00014-08-1-1195), the E&P Sound and Marine Life Joint Industry Project of the
International Association of Oil and Gas Producers (JIP2207-23); the University of California
Natural Reserve System, the NOAA Dr. Nancy Foster Scholarship Program, and the Moore,
Packard, and Sloan Foundations.

LITERATURE CITED

- 1. K. Ralls, Sexual dimorphism in mammals: avian models and unanswered questions. *Am*
*Nat* **111**, 917-938 (1977).
2. J. L. Isaac, Potential causes and life - history consequences of sexual size dimorphism in
mammals. *Mamm Rev* **35**, 101-115 (2005).
3. T. Clutton-Brock, Sexual selection in males and females. *Science* **318**, 1882-1885 (2007).
4. D. E. Promislow, R. Montgomerie, T. E. Martin TE, Mortality costs of sexual dimorphism in
birds. *Proc R Soc B* **250**, 143-150 (1992).
5. L. W. Simmons, D. J. Emlen DJ, Evolutionary trade-off between weapons and testes. *Proc*
*Natl Acad Sci* **103**, 16346-16351 (2006).

6. J. F. Lemaître, *et al.*, Sex differences in adult lifespan and aging rates of mortality across wild
mammals. *Proc Natl Acad Sci* **117**, 8546-8553 (2020).
7. W. U. Blanckenhorn, Behavioral causes and consequences of sexual size
dimorphism. *Ethology* **111**, 977-1016 (2005).
8. M. R. Robinson, J. G. Pilkington, T. H. Clutton-Brock, J. M. Pemberton, L. E. Kruuk, Live
fast, die young: trade-offs between fitness components and sexually antagonistic selection on
weaponry in Soay sheep. *Evol* **60**, 2168-2181 (2006).
9. C. Schradin, L. D. Hayes, A synopsis of long-term field studies of mammals: achievements,
future directions, and some advice. *J Mammal* **98**, 670-677 (2017).
10. R. H. MacArthur, E. R. Pianka, On optimal use of a patchy environment. *Am Nat* **100**, 603-
609 (1966).
11. T. W. Schoener, Theory of feeding strategies. *Annu Rev Ecol Syst* **2**, 369-404 (1971).
12. P. J. Moors, Sexual dimorphism in the body size of mustelids (Carnivora): the roles of food
habits and breeding systems. *Oikos* **34**, 147-158 (1980).
13. R. Shine, Ecological causes for the evolution of sexual dimorphism: a review of the
evidence. *Q Rev Biol* **64**, 419-461 (1989).
14. A. Herrel, L. Spithoven, R. Van Damme, F. D. Vree, Sexual dimorphism of head size in
*Gallotia galloti*: testing the niche divergence hypothesis by functional analyses. *Funct Ecol* **13**,
289-297 (1999).
15. T. W. Schoener, Theory of feeding strategies. *Annu Rev Ecol Syst* **2**, 369-404 (1971).
16. S. L. Lima, P. A. Bednekoff, Temporal variation in danger drives antipredator behavior: the
predation risk allocation hypothesis. *Am Nat* **153**, 649-659 (1999).
17. M. Hebblewhite, E. H. Merrill, Trade-offs between predation risk and forage differ between
migrant strategies in a migratory ungulate. *Ecol* **90**, 3445-3454 (2009).
18. N. C. Bonnot *et al.*, Stick or twist: roe deer adjust their flight behaviour to the perceived
trade-off between risk and reward. *Anim Behav* **124**, 35-46 (2017).
19. R. K. Selander, Sexual dimorphism and differential niche utilization in birds. *Condor* **68**,
113-151 (1966).
20. C. A. Hierlihy, R. Garcia-Collazo, C. B. Chavez Tapia, F. F. Mallory, Sexual dimorphism in
the lizard *Sceloporus siniferus*: support for the intraspecific niche divergence and sexual
selection hypotheses. *Salamandra* **49**, 1-6 (2013).

21. B. J. Le Boeuf, Male-male competition and reproductive success in elephant seals. *Am*
*Zool* **14**, 163-176 (1974).
22. C. J. Deutsch, M. P. Haley, B. J. Le Boeuf, Reproductive effort of male northern elephant
seals: estimates from mass loss. *Can J Zool* **68**, 2580-2593 (1990).
23. G. A. Bartholomew, A model for the evolution of pinniped polygyny. *Evol* **24**, 546-559
(1970).
24. B. J. Le Boeuf, D. E. Crocker, S. B. Blackwell, P. A. Morris, P. H. Thorson, Sex differences
in diving and foraging behaviour of northern elephant seals. *Symp Zool Soc Lond* **66**, 149-178
(1993).
25. J. Reiter, K. J. Panken, B. J. Le Boeuf, Female competition and reproductive success in
northern elephant seals. *Anim Behav* **29**, 670-687 (1981).
26. B. Le Boeuf, R. Condit, J. Reiter, Lifetime reproductive success of northern elephant seals
(*Mirounga angustirostris*). *Can J Zool* **97**, 1203-1217 (2019).
27. R. L. DeLong, B. S. Stewart, Diving patterns of northern elephant seal bulls. *Mar Mamm*
*Sci* **7**, 369-384 (1991).
28. B. S. Stewart, R. L. DeLong, Double migrations of the northern elephant seal, *Mirounga*
*angustirostris*. *J Mammal* **76**, 196-205 (1995).
29. B. J. Le Boeuf, *et al.*, Foraging ecology of northern elephant seals. *Ecol Monogr* **70**, 353-382
(2000).
30. S. E. Simmons, D. E. Crocker, R. M. Kudela, D. P. Costa, Linking foraging behaviour of the
northern elephant seal with oceanography and bathymetry at mesoscales. *Mar Ecol Prog*
*Ser* **346**, 265-275 (2007).
31. P. W. Robinson, S. E. Simmons, D. E. Crocker, D. P. Costa, Measurements of foraging
success in a highly pelagic marine predator, the northern elephant seal. *J Anim Ecol* **79**, 1146-
1156 (2010).
32. P. W. Robinson, *et al.*, Foraging behavior and success of a mesopelagic predator in the
northeast Pacific Ocean: insights from a data-rich species, the northern elephant seal. *PLoS*
*One* **7**, e36728 (2012).
33. S. H. Peterson, J. T. Ackerman, D. P. Costa, Marine foraging ecology influences mercury
bioaccumulation in deep-diving northern elephant seals. *Proc R Soc B* **282**, 20150710 (2015).

34. M. S. Lowry, *et al.*, Abundance, Distribution, and Population Growth of the Northern
Elephant Seal (*Mirounga angustirostris*) in the United States from 1991 to 2010. *Aquat*
*Mamm* **40**, 20-31 (2014).
35. A. F. Henderson, *et al.*, Inferring variation in southern elephant seal at-sea mortality by
modelling tag failure. *Front Mar Sci* **7**, doi: 10.3389/fmars.2020.517901 (2020).
36. G. A. Antonelis Jr, M. S. Lowry, D. P. DeMaster, C. H. Fiscus, Assessing northern elephant
seal feeding habits by stomach lavage. *Mar Mamm Sci* **3**, 308-322 (1987).
37. G. A. Antonelis, M. S. Lowry, C. H. Fiscus, B. S. Stewart, R. L. DeLong, Diet of the
northern elephant seal: in *Elephant Seals: Population Ecology, Behavior, and Physiology*, B. J.
Le Boeuf, R. M. Laws (University of California Press, 1994), pp 211-223.
38. C. Goetsch, *et al.*, Energy-rich mesopelagic fishes revealed as a critical prey resource for a
deep-diving predator using quantitative fatty acid signature analysis. *Front Mar Sci* **5**, 430
(2018).
39. L. A. Hückstädt, R. R. Holser, M. S. Tift, D. P. Costa, The extra burden of motherhood:
reduced dive duration associated with pregnancy status in a deep-diving mammal, the northern
elephant seal. *Biol Letters* **14**, 20170722 (2018).
40. N. E. Humphries, *et al.*, Environmental context explains Lévy and Brownian movement
patterns of marine predators. *Nature* **465**, 1066 (2010).
41. A. M. Sequeira, *et al.*, Convergence of marine megafauna movement patterns in coastal and
open oceans. *Proc Natl Acad Sci* **115**, 3072-3077 (2018).
42. B. A. Bluhm, R. Gradinger. Regional variability in food availability for Arctic marine
mammals. *Ecol Appl* **18**, S77-S96.
43. B. J. McConnell, C. Chambers, M. A. Fedak, Foraging ecology of southern elephant seals in
relation to the bathymetry and productivity of the Southern Ocean. *Antarct Sci* **4**, 393-398
(1992).
44. D. E. Crocker, D. S. Houser, P. M. Webb, Impact of body reserves on energy expenditure,
water flux, and mating success in breeding male northern elephant seals. *Physiol Biochem*
*Zool* **85**, 11-20 (2012).
45. C. R. Cox, Agonistic encounters among male elephant seals: frequency, context, and the role
of female preference. *Am Zool* **21**, 197-209 (1981).

46. K. M. Colegrove, D. J. Greig, F. M. Gulland, Causes of live strandings of northern elephant
seals (*Mirounga angustirostris*) and Pacific harbor seals (*Phoca vitulina*) along the central
California coast, 1992-2001. *Aquat Mamm* **31**, 1-10 (2005).
47. S. J. Jorgensen, *et al*, Killer whales redistribute white shark foraging pressure on
seals. *Scientific Rep* **9**, 1-9 (2019).
48. T. A. Jefferson, P. J. Stacey, R. W. Baird, A review of killer whale interactions with other
marine mammals: Predation to co-existence. *Mammal Rev* **21**, 151-180 (1991).
49. R. W. Baird, L. M. Dill, Occurrence and behaviour of transient killer whales: seasonal and
pod-specific variability, foraging behaviour, and prey handling. *Can J Zool* **73**, 1300-1311
(1995).
50. J. K. Ford, *et al.*, Dietary specialization in two sympatric populations of killer whales
(*Orcinus orca*) in coastal British Columbia and adjacent waters. *Can J Zool* **76**, 1456-1471
(1998).
51. B. J. Le Boeuf, M. Riedman, R. S. Keyes, White shark predation on pinnipeds in California
coastal waters. *Fish Bull* **80**, 891-895 (1982).
52. A. P. Klimley, D. G. Ainley, Great White Sharks: The Biology of *Carcharodon carcharias*
(Academic Press, 1996).
53. J. H. Moxley, *et al*, Non-trophic impacts from white sharks complicate population recovery
for sea otters. *Ecol Evol* **9**, 6378-6388 (2019).
54. R. Condit, *et al.*, Lifetime survival rates and senescence in northern elephant seals. *Mar*
*Mamm Sci* **30**, 122-138 (2014).
55. M. P. Haley, C. J. Deutsch, B. J. Le Boeuf, Size, dominance and copulatory success in male
northern elephant seals, *Mirounga angustirostris*. *Anim Behav* **48**, 1249-1260 (1994).
56. J. González-Solís, J. P. Croxall, A. G. Wood, Sexual dimorphism and sexual segregation in
foraging strategies of northern giant petrels, *Macronectes halli*, during incubation. *Oikos* **90**,
390-398 (2000).
57. M. A. Hindell, D. J. Slip, H. R. Burton, The diving behavior of adult male and female
southern elephant seals, *Mirounga-Leonina* (Pinnipedia, Phocidae). *Aust J Zool* **39**, 595-619
(1991).

58. R. Lewis, T. C. O'Connell, M. Lewis, C. Campagna, A. R. Hoelzel, Sex-specific foraging
strategies and resource partitioning in the southern elephant seal (*Mirounga leonina*). *Proc R Soc*
*Lond B Biol Sci* **273**, 2901-2907 (2006).
59. G. A. Breed, W. D. Bowen, J. I. McMillan, M. L. Leonard, Sexual segregation of seasonal
foraging habitats in a non-migratory marine mammal. *Proc R Soc B* **273**, 2319-2326 (2006).
- 60. D. J. Krause, M. E. Goebel, C. M. Kurle. Leopard seal diets in a rapidly warming polar
region vary by year, season, sex, and body size. *BMC Ecol* **20**, 1-15 (2020).
- 61. P. Lindenfors, B. S. Tullberg, M. Biuw, Phylogenetic analyses of sexual selection and sexual
size dimorphism in pinnipeds. *Behav Ecol Sociobiol* **52**, 188-193 (2002).
- 62. T. M. Cullen, D. Fraser, N. Rybczynski, C. Schröder-Adams, Early evolution of sexual
dimorphism and polygyny in Pinnipedia. *Evol* **68**, 1469-1484 (2014).
- 63. B. Page, J. McKenzie, S. D. Goldsworthy, Inter-sexual differences in New Zealand fur seal
diving behaviour. *Mar Ecol Prog Ser* **304**, 249-264 (2005).
- 64. G. A. Breed, W. D. Bowen, J. I. McMillan, M. L. Leonard, Sexual segregation of seasonal
foraging habitats in a non-migratory marine mammal. *Proc Royal Soc B* **273**, 2319-2326 (2006).
- 65. G. Sheffield, J. M. Grebmeier, Pacific walrus (*Odobenus rosmarus divergens*): differential
prey digestion and diet. *Mar Mamm Sci* **25**, 761-777 (2009).
- 66. C. E. Bajzak, S. D. Côté, M. O. Hammill, G. Stenson G. Intersexual differences in the
postbreeding foraging behaviour of the Northwest Atlantic hooded seal. *Mar Ecol Prog Ser* **385**,
285-294 (2009).
- 67. K. E. Ruckstuhl, P. Neuhaus, Sexual segregation in ungulates: a comparative test of three
hypotheses. *Biol Rev* **77**, 77-96 (2002).
- 68. M. S. Mooring, *et al.*, Sexual segregation in desert bighorn sheep (*Ovis Canadensis*
*mexicana*). *Behaviour* **140**, 183-207 (2003).
- 69. B. J. Le Boeuf, D. P. Costa, A. C. Huntley, S. D. Feldkamp, Continuous, deep diving in
female northern elephant seals, *Mirounga angustirostris*. *Can J Zool* **66**, 446-458 (1988).
- 70. S. E. Simmons, *et al.*, Climate-scale hydrographic features related to foraging success in a
capital breeder, the northern elephant seal *Mirounga angustirostris*. *Endanger Species Res* **10**,
233-243 (2010).
- 71. M. Kleiber, Metabolic turnover rate: a physiological meaning of the metabolic rate per unit
body weight. *J Theor Biol* **53**, 199-204 (1975).

72. G. A. J. Worthy, P. A. Morris, D. P. Costa, B. L. Le Boeuf, Moulting energetics of the northern
elephant seal (*Mirounga angustirostris*). *J Zool* **227**, 257-265 (1992).
- 73. P. M. Webb, D. E. Crocker, S. B. Blackwell, D. P. Costa, B. J. Boeuf, Effects of buoyancy
on the diving behavior of northern elephant seals. *J Exp Biol* **201**, 2349-2358 (1998).
- 10 618 74. D. E. Crocker, J. D. Williams, D. P. Costa, B. J. Le Boeuf, Maternal traits and reproductive
11 619 effort in northern elephant seals. *Ecology* **82**, 3541-3555 (2001).
- 13 620 75. R Core Team, R: A language and environment for statistical computing. R Foundation for
14 621 Statistical Computing, Vienna, Austria. <https://www.R-project.org/> (2017).
- 17 622 76. T. T. Sutton, *et al.*, A global biogeographic classification of the mesopelagic zone. *Deep Sea*
18 623 *Res Part I Oceanogr Res Pap* **126**, 85-102 (2017).
- 20 624 77. H. Hakoyama, B. J. Le Boeuf, Y. Naito, W. Sakamoto. Diving behavior in relation to
21 625 ambient water temperature in northern elephant seals. *Can J Zool* **72**, 643-651 (1994).
- 23 626 78. P. K. Visscher, T. D. Seeley, Foraging strategy of honeybee colonies in a temperate
24 627 deciduous forest. *Ecology* **63**, 1790-1801 (1982).
- 79. M. E. Gilmour, *et al.*, Plasticity of foraging behaviors in response to diverse environmental
conditions. *Ecosphere* **9**, 1-19 (2018).
- 80. S. Lê, J. Josse, F. Husson, FactoMineR: an R package for multivariate analysis. *J Stat*
*Softw* **25**, 1-18 (2008).
- 81. J. Josse, F. Husson, missMDA: a package for handling missing values in multivariate data
analysis. *J Stat Softw* **70**, 1-31 (2016).
- 82. R. V. Lenth, Least-squares means: the R package lsmeans. *J Stat Softw* **69**, 1-33 (2016).
- 83. N. W. Cooper, T. W. Sherry, P. P. Marra, Modeling three-dimensional space use and overlap
in birds. *Auk* **131**, 681-693 (2014).
- 84. C. A. Simpfendorfer, E. M. Olsen, M. R. Heupel, E. Moland E, Three-dimensional kernel
utilization distributions improve estimates of space use in aquatic animals. *Can J Fish Aquat Sci*
**69**, 565-572 (2012).
- 85. J. Fieberg, C. O. Kochanny, Quantifying home-range overlap: the importance of the
utilization distribution. *J Wildl Manag* **69**, 1346-1359 (2005).

642
643
643
644

FIGURE LEGENDS

FIGURE 1. Comparison of satellite tracks and dive behavior of 39 male and 181 female northern
elephant seals (*Mirounga angustirostris*). A) Males travel (blue lines) to coastal areas and forage
(blue circles) on the continental shelf (gray area). Females travel (orange lines) and forage
(orange circles) throughout the North Pacific. The three mesopelagic ecosystems (boundaries
defined by Sutton et al., 76) utilized by northern elephant seals are color-coded and labeled, with
the California Current ecoregion in aqua, the Subarctic Pacific in light blue, and the North
Central Pacific in blue-grey. Northern elephant seal illustration by Pieter Folkens. B) Expanded
view of differential male and female habitat use on/near the continental shelf in the Subarctic
Pacific. C) Representative dive profile of daytime benthic foraging (black line) and pelagic
foraging dives (gray line) from a male and female seal, respectively. The benthic foraging dives
represented here occurred on the continental shelf, and the pelagic foraging dives were adjacent
to the shelf. D) Tukey's boxplots comparing the proportions of the two foraging dive types used
by both sexes on their foraging trips: benthic foraging dives (black) and pelagic foraging dives
(gray). Horizontal bars denote the 25th quartile, 50th (median), and 75th quartile, respectively.

FIGURE 2. Comparison of foraging success and mortality rate metrics between male (blue) and
female (orange) northern elephant seals. Tukey's boxplots compare sex-specific differences in
663 A) mass gain (kg), and B) rate of energy gain (MJ d⁻¹) of 32 males and 128 females. Horizontal
bars represent the 25th quartile, 50th (median), and 75th quartile, respectively. C) Probability of at-
sea mortality for each sex on the foraging trip. Values are model mean \pm 95% confidence
intervals. D) Satellite tracks of males (n=17) and females (n=22) that died at sea. Circles
represent the point of last satellite transmission. The three mesopelagic ecosystems (boundaries
defined by Sutton et al., 76) utilized by northern elephant seals are color-coded and labeled, with
the California Current ecoregion in aqua, the Subarctic Pacific in light blue, and the North
Central Pacific in blue-grey. E) Density plot of dead males and females showing their distance to
the continental shelf at their last satellite transmission. Vertical lines are the mean value for each
sex.

TABLES

**TABLE 1.** Northern elephant seal movement and dive behavior variables (mean±SD) associated
 with the three foraging strategies.

Variable	Males (n=32)	Post-breeding females (n=94)	Post-molt females (n=34)
Distance to continental shelf (km)	22.17±55.73 ^A	509.72±353.15 ^B	569.53±371.59 ^B
Proportion of time spent feeding	0.53±0.16	0.58±0.13	0.58±0.08
Foraging area (km ²)	74,911±203,029 ^A	171,159±413,562 ^A	334,526±464,054 ^B
Proportion of transit dives	0.35±0.13	0.31±0.10	0.28±0.07
Proportion of pelagic foraging dives (PFD)	0.15±0.08 ^A	0.54±0.14 ^B	0.53±0.13 ^B
Proportion of drift dives	0.11±0.04	0.11±0.12	0.13±0.06
Proportion of benthic foraging dives (BFD)	0.40±0.20 ^A	0.04±0.03 ^B	0.07±0.11 ^B
Max depth, day PFD (m)	406.51±83.43 ^A	597.17±53.96 ^{B,*}	605.11±37.76 ^{B,*}
Max depth, night PFD (m)	364.60±84.19 ^A	503.45±35.51 ^{B,*}	494.18±29.65 ^{B,*}
Dive duration, day PFD (min)	23.90±3.63 ^A	25.18±2.13 ^{A,*}	27.98±0.32 ^{B,*}
Dive duration, night PFD (min)	22.75±3.52 ^A	20.42±1.62 ^{B,*}	23.82±1.80 ^{A,*}
Bottom time, day PFD (min)	12.50±1.65 ^A	11.22±1.44 ^{B,*}	13.13±1.92 ^{A,*}
Bottom time, night PFD (min)	11.99±1.41 ^A	9.35±1.18 ^{B,*}	11.99±1.35 ^{A,*}
Post-dive interval, PFD (min)	2.56±0.30 ^A	2.01±0.25 ^B	2.26±0.25 ^C
No. vertical excursions, day PFD	19.54±2.33 ^A	17.50±1.95 ^{B,*}	18.60±2.26 ^{A,*}
No. vertical excursions, night PFD	19.12±2.22 ^A	15.00±1.41 ^{B,*}	17.44±1.91 ^{C,*}
Efficiency, day PFD	0.47±0.04 ^A	0.41±0.03 ^B	0.43±0.04 ^{C,*}
Efficiency, night PFD	0.47±0.04 ^A	0.42±0.03 ^B	0.46±0.03 ^{A,*}
Max depth, day BFD (m)	241.37±104.52 ^A	255.29±113.79 ^{A,*}	477.83±106.40 ^{B,*}
Max depth, night BFD (m)	219.59±87.35 ^A	165.08±50.43 ^{B,*}	307.89±182.10 ^{C,*}
Dive duration, day BFD (min)	20.73±2.95 ^A	19.93±4.27 ^{A,*}	30.08±4.62 ^{B,*}
Dive duration, night BFD (min)	19.92±2.28 ^A	18.20±2.77 ^{A,*}	24.75±2.53 ^{B,*}
Bottom time, day BFD (min)	13.49±1.46 ^{A,*}	12.12±2.90 ^{A,*}	16.03±2.20 ^{B,*}
Bottom time, night BFD (min)	12.06±1.69 ^{A,*}	11.14±2.20 ^{A,*}	13.77±2.39 ^{B,*}
Post-dive interval, day BFD (min)	2.42±0.47 ^A	1.65±0.47 ^{B,*}	2.26±0.64 ^A
Post-dive interval, night BFD (min)	3.43±3.20	4.10±9.70 [*]	2.80±1.07
No. vertical excursions, day BFD	18.20±2.65	20.99±7.73	19.64±5.50 [*]
No. vertical excursions, night BFD	16.18±3.22	19.07±8.03	15.77±4.04 [*]
Efficiency, BFD	17.08±2.98	20.18±7.42	18.05±4.62

Letters show significant differences between strategies from post-hoc pairwise contrasts ($p \leq 0.05$).

* show significant differences between day and night dive variables within a strategy ($p \leq 0.05$).

**TABLE 2.** Comparison of 2D and 3D foraging ranges (95% utilization distribution, UDs) and
 core foraging areas (50% UDs) and percentage of overlap of the foraging ranges and core
 foraging areas between male and female northern elephant seals.

Sex	Kernel Density	2D		3D	
		Area (km ²)	% Overlap	Area (km ³)	% Overlap
Male	95%	188	9.92	51,509	21.6
	50%	42	0	447	3.88
Female	95%	463	4.03	221,876	5.01
	50%	93.3	0	278	6.25

**TABLE 3.** Northern elephant seal foraging success variables (mean±SD) for the three foraging
 strategies.

Variable	Males (n=32)	Post-breeding females (n=94)	Post-molt females (n=34)
Departure body mass (kg)	1,060.70±182.47 ^A	318.27±43.61 ^B	297.79±48.21 ^C
Mass gain on trip (kg)	458.38±218.26 ^A	96.43±59.93 ^B	227.49±67.30 ^C
Mass gain rate on trip (kg day ⁻¹)	3.63±1.61 ^A	0.96±0.27 ^B	1.13±0.22 ^B
Mass gain rate relative to feeding time (kg day ⁻¹)	5.66±4.40 ^A	1.89±0.89 ^B	2.03±0.78 ^B
Proportion of mass gain on trip	0.44±0.23 ^A	0.32±0.25 ^A	0.80±0.26 ^B
Energy gain (MJ)	8,020±2,085 ^A	1,638±918 ^B	3,794±1,052 ^C
Energy gain rate on trip (MJ day ⁻¹)	67.31±23.10 ^A	17.28±6.67 ^B	18.93±4.77 ^B
Energy gain rate relative to feeding time (MJ day ⁻¹)	98.46±66.71 ^A	30.97±13.36 ^B	31.37±12.32 ^B

Letters indicate significant differences between strategies based on post-hoc pairwise contrasts ($p \leq 0.05$).

686

FIGURE 1. Comparison of satellite tracks and dive behavior of 39 male and 181 female northern elephant seals (*Mirounga angustirostris*). A) Males travel (blue lines) to coastal areas and forage (blue circles) on the continental shelf (gray area). Females travel (orange lines) and forage (orange circles) throughout the North Pacific. The three mesopelagic ecosystems (boundaries defined by Sutton et al., 76) utilized by northern elephant seals are color-coded and labeled, with the California Current ecoregion in aqua, the Subarctic Pacific in light blue, and the North Central Pacific in blue-grey. Northern elephant seal illustration by Pieter Folkens. B) Expanded view of differential male and female habitat use on/near the continental shelf in the Subarctic Pacific. C) Representative dive profile of daytime benthic foraging (black line) and pelagic foraging dives (gray line) from a male and female seal, respectively. The benthic foraging dives represented here occurred on the continental shelf, and the pelagic foraging dives were adjacent to the shelf. D) Tukey's boxplots comparing the proportions of the two foraging dive types used by both sexes on their foraging trips: benthic foraging dives (black) and pelagic foraging dives (gray). Horizontal bars denote the 25th quartile, 50th (median), and 75th quartile, respectively.

FIGURE 2. Comparison of foraging success and mortality rate metrics between male (blue) and female (orange) northern elephant seals. Tukey's boxplots compare sex-specific differences in A) mass gain (kg), and B) rate of energy gain (MJ d⁻¹) of 32 males and 128 females. Horizontal bars represent the 25th quartile, 50th (median), and 75th quartile, respectively. C) Probability of at-sea mortality for each sex on the foraging trip. Values are model mean \pm 95% confidence intervals. D) Satellite tracks of males (n=17) and females (n=22) that died at sea. Circles represent the point of last satellite transmission. The three mesopelagic ecosystems (boundaries defined by Sutton et al., 76) utilized by northern elephant seals are color-coded and labeled, with the California Current ecoregion in aqua, the Subarctic Pacific in light blue, and the North Central Pacific in blue-grey. E) Density plot of dead males and females showing their distance to the continental shelf at their last satellite transmission. Vertical lines are the mean value for each sex.

Appendix B

Baylor University

DEPARTMENT OF BIOLOGY
B.207 BAYLOR SCIENCE BUILDING
ONE BEAR PLACE #97388 * WACO, TX 76798

October 29, 2021

Dear RSOS Editor and Reviewers,

My coauthors and I would like to resubmit our manuscript titled “Risk reward trade-offs drive sex-specific foraging strategies in sexually dimorphic northern elephant seals” for publication in the Royal Society Open Science.

Thank you for your response to our submission and for your thoughtful letter, along with the detailed comments and suggestions of the two expert reviewers. We have carefully considered each comment provided, and we have edited the manuscript to make the suggested improvements and clarifications. My coauthors and I feel that the manuscript has been strengthened by this process. We have included specific responses to all of the reviewers’ comments below.

We look forward to your response to our revised manuscript and hope that you find it suitable for publication in Royal Society Open Science.

Sincerely,

Sarah Kienle, Ph.D.

Assistant Professor

Department of Biology

Baylor University

RESPONSE TO REVIEWS

SUBJECT EDITOR

1. We have changed the title so that ‘mammal’ is now ‘northern elephant seals’ as suggested.

ASSOCIATE EDITOR, DR. GREIG

1. Based on Reviewer 2’s comment, we modified the text, making sure to specify that we are referring to mortality risk differences between male and female foraging strategies rather than overall risk (which is outside the scope of the study).

REVIEWER 1

General note: Thank you for your detailed comments – it was really helpful to see areas where we were unintentionally confusing or had left out enough detail. We really appreciate your attention to detail and the feedback, especially on the reporting of our results and methods. Thank you!

General Comments

1. Split of male/female and female/season strategies: We added text to the beginning of the results (127-130) explaining that the split between the three strategies (males vs. female post breeding & female post-molt) was based on the results of our hierarchical clustering analysis. We also added a clarifying sentence in the paragraph in the results about male feeding strategies not differing between the trips (lines 142-145).
2. HCA results: The HCA results are more explicitly referred to in the text (127-130), and the results are presented in a supplementary table (Table S3) that shows the cluster assignments for each individual seal.
3. Reporting results. We appreciate Reviewer 1’s comment pointing out that we switched between ranges and actual values throughout the results– and we appreciate that this was—indeed—confusing. As a result, we have eliminated ranges and now only report the mean values \pm the standard deviation.
4. Mortality/survey effort. We expanded the methods section to provide details on how individual seals were identified – in addition to the deployment and tracking of the satellite tags (351-357). We also included details on the mark/recapture efforts and daily/weekly resights and annual censuses that provide the longitudinal data on mortality used in this study and sources of bias in the estimates (449-460).
5. Literature cited: Thank you for the suggestion of the additional references. Both are extremely valuable and relevant to this paper. We have added these references throughout the manuscript – particularly in the discussion.

Specific Comments

1. Line 127: We added text describing the breakdown of the HCA results into three clusters – the male strategy and the two female strategies (lines 127-130). We had a nearly even split of males tracked on the post-breeding and post-molt trips, but there were no statistical differences between the post-breeding and post-molt foraging trips. We added a description of this in (now) lines 142-145. And, we report the HCA results in a supplementary table (Table S1-S3).
2. Line 130 & Table 1: The split between trips for females and the lack of split for males was the result of the HCA, and we have added text to address this point (lines 127-130, 142-145).
3. Line 135-136: We added the comparable means and standard deviations for the female strategies to these sentences (and throughout the results).
4. Line 136-137 (now 142-145): We modified this sentence to address the comment above why males were not subdivided into seasonally specific strategies.
5. Line 140 (now 147): We removed the values from this sentence since they are all written out in the previous paragraph.
6. Line 144, 155-157 (now paragraph starting on line 146): These values do not represent the range (that would, indeed, be suspicious). We were trying to show the range in mean values between foraging strategies, which, we now see, is unclear and confusing. We have changed these sentences (and subsequent ones) to report the means and standard deviations for each female foraging trip. We did this for both the dive depth and duration.
7. Line 177-179 (now paragraph starting on line 198): We removed the statistics from this section as suggested and only statistically compared the proportions of mass gain.
8. Line 217 (now 240): We have clarified our description of the male foraging strategy (now lines 142-145) to better describe how male are at sea twice a year on post-breeding and post-molt foraging trips, but that their foraging behavior on these two trips do not differ (in comparison to the female trips that do).
9. Line 291 (now 315): We have added more recent references to this section, as suggested. Thank you.
10. Line 391-392 (now paragraph starting on 444): We expanded multiple parts of the methods section to provide details on census and resighting efforts, as well as our flipper tagging procedures (and we made sure to explain that all instrumented seals had flipper tags; lines 351-357, lines 454-458).
11. Line 423 (now 496): Yes! We love this paper. It was a complete mistake on our end that it wasn't in our literature cited already. It is added now. Thank you.
12. Table 1. We have added text to the beginning of the results section (now lines 127-130), as well as to the table legend. These three foraging strategies were identified from hierarchical clustering analysis of movement patterns and dive behavior – they were not assigned a priori. We added the phrase (unitless) in the title of the dive efficiency variables, which are unitless ratios.
13. Figure 1A. We have added a north arrow, reoriented the map so that it points directly up, and added coordinates.
14. Figure 1B. We have added a north arrow, reoriented the map so that it points directly up, and added coordinates.
15. Figure 1 legend. We have removed 'Tukey's' from our description of the boxplots in the figure legend.

16. Figure 2 legend. We have removed ‘Tukey’s from our description of the boxplots in the figure legend.
17. Figure 2 legend. We removed the statistics from the text on foraging success as suggested above. We still include the comparisons in our figures for readers that do not know as much about elephant seals. We feel that by including the boxplots here, we help visually show the trade-off between mass/energy gain and mortality rate between the sexes.

REVIEWER 2

General note: Thank you for taking the time to look at the data, code, and reproducibility of the data. I’ve never had any detailed feedback on it before, so this was *so* helpful. I really feel that the manuscript – as well as my ability to make sure that my data are truly reproducible – have benefited from this process from your helpful and specific comments on the code and statistical analyses. *Thank you.*

General Comments

1. Methodological details. We have greatly expanded the methods section so that we no longer gloss over the methodological details and are clear about the decisions that went into analyzing these data. We have also added some supplemental tables that provide the output of several of our analyses (PCA, HCA, and model testing).
2. Reproducibility. We did a lot of work to make sure that our work is reproducible. In short, we edited our R script files and csv files to make a logical order and flow between scripts and csv files. We created an informative naming scheme so that it is clear what files are associated with which scripts to be able to reproduce the analyses. We also regrouped the analyses and scripts to follow the logic/ sequence of methods and analyses laid out in the manuscript. All data is now clearly labeled and able to be run without having any guesswork about where data go. Within each R script, we have commented out the workflow, with explanations of our rationale and methodology.
3. Model checking. We have added model checking to all of our analyses, applied a Bonferroni correction to our t-test p-values, and applied different significance criteria appropriately.

Specific Comments

1. Foraging success. We used direct measurements of body composition (body lengths, girths, blubber thickness, mass) and estimates of mass/energy gain & rates of mass/energy gain to calculate foraging success for individual seals. This approach is similar to Robinson et al. (2010) which used the same body composition metrics to validate using drift rate as a measure of foraging success. Because we had actual measures of body composition and could measure/calculate mass and energy gain, we did not use drift rate (or any dive behavior) as our indicator of foraging success. We define our metrics of foraging success as increased relative rates of mass and energy gain (now line 198). In the methods, we describe

how body composition data were collected (paragraph starting on lines 361) and our foraging success metrics (paragraph starting on line 429).

2. Assuming reader knowledge of elephant seals. We addressed Reviewer 2's specific comments and combed through the results to explain any confusing terms that aren't explained until later on in the methods. For example, we explain the evidence used to infer feeding (now line 133-134), the difference between trips where seals were focused vs. feeding throughout (now lines 136-137 and lines 168-169), the relevance of number of vertical excursions to foraging (lines 158-159), and dive efficiency (now lines 178-179).
3. Line 218-219 (now 241-242): By providing context for different variables (based on suggestion #2 above), we now have added text explaining that vertical excursions are a proxy for prey capture attempts in both the results (line 158-159) and methods (now lines 419-420).
4. Line 235 (now 259-260): We cannot be sure that males do not opportunistically feed during transit, and we modified our sentence to better represent our findings. We also detail how feeding events were determined in the methods (443-437).
5. Line 271 (now 294): We edited the sentence (and the manuscript as a whole) to specify 'the risk of mortality', as opposed to 'overall risk', which more accurately represents our results. We also searched the manuscript and made the same edit throughout so that we do not overstate our findings about risk since there are many variables that we cannot control/test here.
6. Line 320-323: We expanded our methods to provide more information on data collection, methodologies, and references. For example, we provide more details on our satellite tag processing (lines 380-390) and processing the TDR data (390-400). We did not use any SMRU tags for dive analysis, so we did not use the broken stick algorithm. We only used archived tracking data recovered from the MK9 or MK10 tags. All tags were subsampled to 8 seconds (and these details have been added to the methods).
7. Line 352 (now 390): We have added citations for all referenced packages throughout the methods. For example, the HCA was conducted using the *cluster* and *factoextra* packages (citations now included). We also expanded the *Data Analysis* section in the methods to describe the method and software used to determine and define dive shape (now paragraph starting on 470).
8. Figure 2D. We modified our language throughout regarding the difference in mortality rate among the different foraging habitats, as suggested. We emphasize what we find – that there are differences in mortality rate between the sexes—but no longer make the assumption that this is due to predation risk or only to foraging in different habitats.

Data and code accessibility

All codes and files have been updated in Dryad.

1. Accessibility

In terms of data organization and accessibility, we have given each input file (CSV) file an easy-to-interpret file name (ex. PCAinput.csv). We also made the output file names more informative

(ex. HCAoutput.csv). We also added text within each R script that details the exact input file required to run each section of code.

The workflow is now as follows, and these steps are included on Dryad as a README file, along with our data and scripts.

Step 1: PCA & HCA

i. PCA

1. Open Kienle_RSOS_PCA and HCA_10-12-2021 (R script)
2. Input: Kienle_RSOS_PCAinput (csv)

ii. HCA

1. Input: Kienle_RSOS_PCAoutput (csv)

Step 2: t-tests

2. Open: Kienle_RSOS_t-tests_10-21-2021 (R script)
3. Input: Kienle_RSOS_t-testsinput (csv)

Step 3: Generalized linear models

4. Open: Kienle_RSOS_GeneralizedLinearModels_10-25-2021 (R script)
5. Input 1: Kienle_RSOS_PCAinput (csv)

Step 4: Spatial Analysis

iii. 3D KD & UD

1. Open: Kienle_RSOS_SpatialAnalysis_10-29-2021 (R script)
2. Input: Kienle_RSOS_SpatialAnalysis_3DFemaleinput (csv)
3. Input: Kienle_RSOS_SpatialAnalysis_3DMaleinput (csv)

iv. 2D KD & UD

1. Input: Kienle_RSOS_SpatialAnalysis_2DFemaleinput (csv)
2. Input: Kienle_RSOS_SpatialAnalysis_2DMaleinput (csv)

Step 5: Mortality Analysis

- v. Open: Kienle_RSOS_MortalityAnalyses_10-19-2021 (R script)
- vi. Input 1: Kienle_RSOS_MortalityAnalysis_MannWhitneyWilcoxontestsinput (csv)
- vii. Input 2: Kienle_RSOS_MortalityAnalysis_glminput (csv)

2. Script: Mortality Analysis

As mentioned earlier, we have moved around the analyses and expanded upon them. Here, in the Mortality analyses, we do the following things:

- 1) We examined the effect of quantitative response variables (transit rate, distance to continental shelf) on mortality rate using non-parametric Mann-Whitney-Wilcoxon tests in R with the Kienle_RSOS_MortalityAnalysis_MannWhitneyWilcoxontests.csv.

- 2) We then fit a binomial GLM to the mortality/survival dataset using the input file: Kienle_RSOS_MortalityAnalysis_glminput.csv file. We expanded our analysis to include multiple models. Variables in the full model include sex, individual, tagging year, and foraging trip. We ran all possible combinations of the model that included sex as the explanatory variable. We then used likelihood ratio tests to compare null and residual deviance and ranked models using the Akaike information criterion. We tested the difference between our best-fitting model and the observed data using a Hosmer-Lemeshow Goodness of Fit test and ANOVA to evaluate model fit and determine the best explanatory variables. We then model the probability of dying by sex and create a figure of our results.

This information on model variables, comparisons, and fit are now included in detail in the methods; we also a table of our model comparison results is included as Table S4 in the supplemental material.

3. Script: t-tests

The t-tests in the t-tests file (Kienle_RSOS_t-testsinput) comparing day/night patterns in dive behavior within each of the three clusters. We examined the distribution of our data for deviations from normality through visual observations (density and Q-Q plots) and normality testing with a Shapiro-Wilks test. For data that followed a normal distribution, we assessed variance through F-tests. Some data had unequal variances between groups, so we used Welch two-sample t-tests to test for differences between time of day on dive behavior. When the assumption of normality was violated, we used non-parametric Mann-Whitney-Wilcoxon tests. Finally, we applied a Bonferroni correction to account for the multiple comparisons and adjusted p-values were used to assess significance. These descriptions have been added to the methods, and the results have been revised accordingly.

4. Script: Generalized linear models

We modified our linear models and added model testing and comparisons of model fit. In brief, we set up generalized linear models to determine which foraging variables significantly differed between clusters using the PCAinput dataset. Each candidate model was compared with a null model (intercept only) using likelihood ratio tests of the null and residual deviances. Models were ranked using the Akaike information criterion corrected for small sample sizes and evaluated model fit using ANOVAs and used estimated marginal means to perform post-hoc pairwise contrasts between each cluster and used Tukey's method for adjusting the p-value for multiple comparisons.

Appendix C

Baylor University

DEPARTMENT OF BIOLOGY
B.207 BAYLOR SCIENCE BUILDING
ONE BEAR PLACE #97388 * WACO, TX 76798

December 13, 2021

Dear RSOS Editor and Reviewers,

My coauthors and I would like to resubmit our manuscript titled “Risk reward trade-offs drive sex-specific foraging strategies in sexually dimorphic northern elephant seals” for publication in the Royal Society Open Science.

Thank you for your response to our submission and for your thoughtful letter. We have further edited the manuscript to make the suggested improvements and clarifications. My coauthors and I agree that the manuscript has been strengthened by this process. We have included specific responses to all of the reviewers’ comments below.

We look forward to your response to our revised manuscript and hope that you find it suitable for publication in Royal Society Open Science.

Sincerely,

Sarah Kienle, Ph.D.

Assistant Professor

Department of Biology

Baylor University

RESPONSE TO REVIEWS

ASSOCIATE EDITOR, DR. GREIG

1. Distance to the continental shelf. We changed our term ‘distance to the continental shelf’ to ‘distance to the continental shelf edge’ throughout the results and discussion as suggested (e.g., see lines 132-133, 146-147, Table 1). We agree that this makes the text much clearer. We also made sure that the text also reflected whether seals were ‘on/over’ the continental shelf (like most males when feeding) or ‘near’ or ‘far’ from the continental shelf edge throughout the discussion. We include a more specific definition of the continental shelf edge in the methods (413-413). Thank you for pointing out a potential source of confusion.
2. Foraging success. You are correct in everything you said about foraging success. Males need absolutely more mass and energy because of their larger body size (which likely drives the sex-specific foraging patterns). But, post-molt females who are pregnant have the highest proportional mass gain due to the costs associated with gestation. In the results (lines 197-208), we take out the definitive statement that males have higher foraging success. Rather, we now emphasize that males have higher absolute mass and energy gain. We also emphasize that post-molt females have the highest proportion of mass gain. In the discussion, we added a bit to our existing text specifying that post-molt females are the ones experiencing the physiological demands of gestation (line 253). We also have text (paragraph beginning on 274), that describes how, males, as the larger sex, need absolutely more energy than females to support and sustain their body masses and this is likely why they feed in benthic continental shelf habitats. Finally, we added some clarifying text to the methods (paragraph beginning on 444) emphasizing the different metrics (absolute vs. relative) used to define foraging success.
3. Line 385 (now 397): We changed examines to examined.
4. Line 386 (now 398): We changed replaces to replaced.

REVIEWER 1

1. We modified the text in the foraging success paragraph of the methods (line 454) to clarify that we retained foraging locations that were associated with a foraging dive type (either pelagic and/or benthic foraging dives). Thanks for identifying this potential source of confusion for readers.